# The effect of climate change on the simulated streamflow of six Canadian rivers based on the CanRCM4 regional climate model

Vivek. K. Arora[1], Aranildo Lima[1], Rajesh Shrestha[2]

[1]Canadian Centre for Climate Modelling and Analysis, Climate Research Division, Environment Canada, Victoria, BC, Canada
[2]Climate Research Division, Environment and Climate Change Canada, Victoria, BC, Canada

2      *Correspondence to*: Vivek K. Arora (vivek.arora@ec.gc.ca)

**Abstract**

The effect of climate change on the hydro-climatology, in particular streamflow, of six major Canadian rivers (Mackenzie, Yukon, Columbia, Fraser, Nelson, and St. Lawrence) is investigated by analyzing results from the historical and future simulations (RCP 4.5 and 8.5 scenarios) performed with the Canadian regional climate model (CanRCM4). Streamflow is obtained by routing runoff using river networks at $0.5°$ resolution. Of these six rivers, Nelson and St. Lawrence are the most regulated. As a result, the streamflow at the mouth of these rivers shows very little seasonality. Additionally, the Great Lakes significantly dampen the seasonality of streamflow for the St. Lawrence River. Mean annual precipitation (P), evaporation (E), runoff (R), and temperature increase for all six river basins in both future scenarios considered here, and the increases are higher for the more fossil fuel-intensive RCP 8.5 scenario. The only exception is the Nelson River basin for which the simulated runoff increases are extremely small. The hydrological response of these rivers to climate warming is characterized by their existing climate states. The northerly Mackenzie and Yukon River basins show a decrease in evaporation ratio (E/P) and an increase in runoff ratio (R/P) since the increase in precipitation is more than enough to offset the increase in evaporation associated with increasing temperature. For the southerly Fraser and Columbia River basins, the E/P ratio increases despite an increase in precipitation, and the R/P ratio decreases due to an already milder climate in the Pacific north-western region. The seasonality of simulated monthly streamflow is also more affected for the southerly Fraser and Columbia Rivers than for the northerly Mackenzie and Yukon Rivers as snow amounts decrease and snowmelt occurs earlier. The streamflow seasonality for the Mackenzie and Yukon rivers is still dominated by snowmelt at the end of the century even in the RCP 8.5 scenario. The simulated streamflow regime for the Fraser and Columbia Rivers shifts from a snow-dominated to a hybrid/rainfall-dominated regime towards the end of this century in the RCP 8.5 scenario. While we expect the climate change signal from CanRCM4 to be higher than other climate models, owing to the higher-than-average climate sensitivity of its parent global climate model, the results presented here provide a consistent overview of hydrological changes across six major Canadian river basins in response to a warmer climate.

## 1. Introduction

As the global population and the standard of living increases so does the strain on freshwater resources. The natural availability of water is determined by the balance between precipitation (P) and evaporation (E) (this includes both evaporation and transpiration from plants). When precipitation exceeds evaporation, which is determined primarily by available energy, the water that does not evaporate or transpire (either at the surface or after infiltration into the soil) termed runoff (R) is carried by the rivers to the oceans. The seasonality of precipitation, its partitioning into snow and rainfall, and the seasonality of snowmelt and evaporation, all of which are determined by the climate in a given catchment or river basin eventually determine the seasonality of runoff. As anthropogenic climate change progresses, changes in the mean annual amounts and the seasonality of these different water budget components will lead to corresponding changes in runoff (Trenberth et al., 2007). Changes in precipitation extremes are also expected to lead to corresponding changes in the extremes of streamflow. The changes in streamflow have implications for floods and power generation. While runoff is expressed in similar units to precipitation and evaporation (depth of water per unit time, e.g. mm/s or m/year), streamflow is the volume of water generated per unit time (e.g. m$^3$/s or km$^3$/year) and requires multiplication with the area over which runoff is generated. Streamflow is also routed down the river network which introduces a time lag and attenuation of the peak runoff.

Output from climate and Earth system models (ESMs) remains the primary source of information for evaluating climate change impacts. Current approaches that rely on information generated by ESMs, to obtain an estimate of how future streamflow may potentially change, may

be classified into two broad categories. The first approach uses simulated runoff directly from
the land surface component of single or multiple climate models which may be routed
downstream to obtain streamflow at the mouths of river basins and at different points along a
given river network (Arora and Boer, 2001; Miller and Russell, 1992; Zhang et al., 2014). Using
direct runoff output from climate models has the benefit that the calculated changes in runoff
are physically consistent with the altered radiative balance of the Earth in response to increases
in the concentrations of greenhouse gases (GHGs). The corresponding changes in the general
circulation of the atmosphere result in the associated changes in near-surface temperature,
precipitation, and the hydrological cycle.  However, this approach suffers from three limitations
– 1) the biases in the climate simulated by the climate model, 2) the fact that the land surface
components of climate models are not calibrated for a given river basin but rather designed to
operate in a reasonably realistic way over the whole globe, and 3) the coarse resolution of global
climate models (GCMs). The last limitation is partially addressed when data from finer-resolution
regional climate models is used. The biases in the simulated climate do affect the simulated
runoff for the current climate. Despite this, the approach can effectively capture the effects of
climate change including increased evaporative demand (Winter and Eltahir, 2012), reduced
snowpack (Salathé et al., 2010; Shrestha et al., 2021a), increased winter streamflow, and earlier
snowmelt-driven peak flow (L. Sushama et al., 2006; Poitras et al., 2011). The second approach
attempts to overcome these limitations by downscaling and/or bias-correcting climate from
climate models for future scenarios and uses that to drive a well-calibrated hydrological model
for given catchments or river basins (Gosling et al., 2011; Ismail et al., 2020; Miller et al., 2021;
Yoosefdoost et al., 2022). The second approach is more prevalent for watershed to regional scale
impacts and adaptation studies. Given the large effort involved in downscaling and bias-
correcting raw climate data from climate models, most current impact studies use downscaled
and bias-corrected data put together by other groups rather than specifically doing this for their
project. Recent examples include the downscaled and bias-corrected climate data for the
conterminous United States (Thrasher et al., 2013) based on climate model output from the fifth
phase of the Coupled Model Intercomparison Project (CMIP5), and statistically downscaled and
bias-corrected data from five CMIP5 models, available at the global scale, tailored to the
requirements of the Inter-Sectoral Impact Model Intercomparison Project (ISIMIP) (Lange, 2019).
Both these data sets have found large applications in the impacts and adaptation community.
The processes of downscaling and bias correction are distinct, and they both have their inherent
limitations. There are several examples of the limited ability of bias-correction to correct and
downscale variability, and that bias-correction can potentially cause implausible climate change
signals (Maraun, 2016; Maraun et al., 2017). There are also uncertainties, substantial
contradictions, and sensitivity to assumptions between the different downscaling methods
(Hewitson et al., 2014).
Finally, while land surface models are typically used within the coupled framework of
climate models, hydrological models are typically used as a standalone model for impact studies.
While the primary output quantities from hydrological models are runoff and streamflow, land
surface models output a range of water, energy, and $CO_2$ fluxes (Blyth et al., 2021; Fisher and
Koven, 2020). The layer of air directly above the land surface, commonly referred to as the
atmospheric or planetary boundary layer, is affected by surface-atmosphere exchanges of energy
and water and extends upward into the atmosphere. A realistic representation of turbulent fluxes
of energy and water in the planetary boundary layer is essential to the transport of moisture and
energy through the atmosphere. As a result, while calibration of hydrological models to
reproduce observed streamflow is a routine exercise (Chegwidden et al., 2019; Hattermann et
al., 2018; Huang et al., 2020; Hundecha et al., 2020), land surface models cannot be calibrated to
reproduce a single or a small subset of quantities. This aspect of land surface versus hydrological
models is also addressed briefly in Bolaños Chavarría et al. (2022). A review by Overgaard et al.
(2006) also attempts to differentiate land surface models from hydrological models. In contrast
to hydrological models, land surface models are expected to reproduce reasonably realistic
estimates of a range of energy, water, and $CO_2$ fluxes over the whole globe. The philosophy
behind land surface models, as they are used in the context of climate models, is that given 1) a
model's structure and parameterizations, 2) the driving geophysical data for fields such as
vegetation cover, soil depth, and soil texture, and 3) the driving meteorological variables, a model
is expected to reasonably realistically reproduce various components of the water, energy, and
carbon cycle at the global scale. The global scale of land surface models within the framework of
climate models precludes tuning of their parameters for individual grid cells or for a region (e.g.
a river basin) to reproduce a small subset of model outputs.

While well-calibrated hydrological models are generally suitable for a given catchment or

a river basin their application cannot be easily extended to large-scale global or regional
hydrologic modelling studies since it is typically not feasible to tune model parameters for all grid
cells in a large domain. For a large region like Canada correctly representing anthropogenic
regulation using downscaled and bias-corrected climate data from an ensemble of climate
models is a challenging task. As a result, this has been done for only a few selected river basins,
considering only one basin at a time. In the end, both approaches have their strengths and
limitations for assessing climate change impacts on hydrology and can be considered
complementary to each other.

Future hydrologic projections using the second approach (hydrological modes driven by

statistically downscaled and bias-adjusted climate models) are available for selected river basins
in Canada. The results over the Prairies and British Columbia (Shrestha et al., 2021b; Sobie and
Murdock, 2022) generally indicate shorter snow cover duration, earlier snowmelt, and reduced
annual maximum snow water equivalent as the climate warms. Streamflow projections across
Canada generally indicate earlier snowmelt-driven peak flow, increased winter flow, and
decreased summer flow (Budhathoki et al., 2022; Dibike et al., 2021; Islam et al., 2019;
MacDonald et al., 2018; Shrestha et al., 2019). Annual streamflow is projected to increase, with
higher increases in the northern basins (Bonsal et al., 2020; Stadnyk et al., 2021). However, these
projections are based on different climate and hydrological models, downscaling methods,
emissions scenarios, and future periods, and no consistent set of projections is available across
all major river basins of Canada.

In this study, we have used the first approach to provide a consistent set of projections

across all major river basins of Canada, while being cognizant of its limitations. We investigate
the effect of climate change on the annual, monthly, and daily streamflow characteristics of six
major Canadian rivers (Mackenzie, Yukon, Columbia, Fraser, Nelson, and St. Lawrence) using
runoff output from simulations performed with version 4 of the Canadian Regional Climate
Model (CanRCM4) (Scinocca et al. 2016). The river basins of the Yukon and Columbia Rivers cover
part of the United States of America as well. We used daily runoff generated from CanRCM4 for
the historical period and for the two future scenarios (representative concentration pathways
(RCP) 4.5 and 8.5). The spatial resolution of runoff data from CanRCM4 is 0.22° which is
equivalent to about 12 km at 60° N (Canada lies between approximately 42°N and 83°N). We
then routed this runoff through river networks at 0.5° resolution to evaluate streamflow at the
mouths of major Canadian rivers. The Mackenzie, Yukon, and Fraser Rivers are somewhat less
regulated than the heavily regulated Nelson, Columbia, and St. Lawrence Rivers. The routing
scheme used here does not take into account dams and reservoirs and therefore the modelled
streamflow represents natural streamflow. This aspect is discussed in more detail in Section 2.
**2. Models and data**

Equation (1) summarizes the water balance over a given grid cell or a river basin for a

given timescale.

$$P = E + R + \Delta S \tag{1}$$

where $\Delta S$ is the change in water storage including that in soil moisture, snow, and the canopy
water storage. All terms are expressed in depth per unit time units (e.g. mm/year). When a
system is in equilibrium, at annual or longer timescales $\Delta S = 0$ and $P = E + R$. $\Delta S$, however,
may not be zero even over long timescales when a system is not in equilibrium e.g., when snow
is accumulating or is melting consistently. We evaluated the P, E, and R components of equation
(1) simulated by CanRCM4 for each of the six river basins, considered in this analysis, and routed
R to obtain streamflow at the river mouths.

**2.1 The Canadian Regional Climate Model  (CanRCM4)**
CanRCM4 uses the fourth-generation Canadian atmospheric physics (CanAM4) package
(von Salzen et al., 2013), which is the product of a multi-decadal program of climate model
development at the Canadian Centre for Climate Modelling and Analysis (CCCma), a section
within Environment and Climate Change Canada. The CanAM4 atmospheric physics package is
also used in CanESM2 (Arora et al., 2011) which contributed results to CMIP5. The difference
between CanRCM4 and CanESM2, other than the former being a regional climate model and the
latter being a comprehensive global ESM, is that CanRCM4 employs the limited-area
configuration of the Global Environmental Multiscale (GEM) model (Côté et al., 1998), which uses
a semi-Lagrangian dynamical core for advection in the atmosphere and is developed by
Environment and Climate Change Canada's Recherche en Prévision Numérique (RPN) where it is
used both for global and regional numerical weather prediction. CanESM2 on the other hand
uses a spectral dynamical core for advection in the atmosphere. CanRCM4 is driven at its
boundaries with data from its parent model (CanESM2). An overview and technical details of the
coordinated global and regional climate modelling effort used to develop the CanESM2-CanRCM4
system are described in detail by Scinocca et al. (2016). Results from the model's North American
0.22° domain, for a single ensemble member, are primarily used here. In addition, we also used
runoff from CanRCM4 0.44° resolution simulations for the North American domain because of
the availability of a large ensemble (LE) of 50 members (CanRCM4 LE) (ECCC, 2018). The large
ensemble simulations allow the consideration of CanRCM4's internal variability, which is an
intrinsic property of the climate system and models, that is largely irreducible and could account
for a large fraction of the inter-climate model spread (Deser et al., 2020). The results used here
from CanRCM4 form part of its contribution to the coordinated regional climate downscaling
experiment (CORDEX) effort. The North American domain of CanRCM uses a rotated latitude-
longitude projection with the North Pole at longitude 83° E and latitude 42.5° N, as opposed to
the geographic North Pole (longitude 0°E, latitude 90° N).

The land surface component in CanAM4 is the coupled CLASS-CTEM model. The physical

processes are based on the Canadian Land Surface Scheme (CLASS) (Verseghy, 1991; Verseghy et
al., 1993), and biogeochemical processes (which simulate vegetation as a dynamic component of
the climate system) are based on the Canadian Terrestrial Ecosystem Model (CTEM) (Arora and
Boer, 2003, 2005). The configuration of CLASS-CTEM used in CanESM2 and CanRCM4 uses three
soil layers with thicknesses of 0.10, 0.25, and 3.75 m. Liquid and frozen soil moisture contents,
and soil temperature, are determined prognostically for the three soil layers. The temperature,
albedo, mass, and density of a single-layer snow pack (when environmental conditions permit
snow to exist) are also prognostically modelled. Surface runoff is generated in CLASS when
precipitation intensity exceeds infiltration capacity and when the top soil layer is saturated. The
rainwater and snow melt that infiltrate the soil are available for soil evaporation and
transpiration. Any remaining water percolates down the soil profile and comes out at the bottom
of the soil profile and is termed drainage. Combined surface runoff and drainage constitute total
runoff. Like most land surface components of ESMs, CLASS does not include a groundwater
representation. Surface runoff and drainage from CLASS are used as input into a large-scale river
routing scheme to route runoff and obtain streamflow at the mouth of the rivers considered in
this study as explained in the next section.
**2.2 Variable velocity routing model**
The variable velocity river routing scheme of Arora and Boer (1999) that is implemented
in the family of Canadian ESMs (CanESMs) (Arora et al., 2009, 2011; Swart et al., 2019) is used to
route daily runoff from CanRCM4. This routing scheme has been implemented in various versions
of CanESMs at a spatial resolution of $2.81°$ since the year 2000. For this study, the routing scheme
was implemented at a spatial resolution of $0.5°$. The reason for using river routing at $0.5°$
resolution instead of scaling river networks to the $0.22°$ rotated latitude-longitude projection of
CanRCM4 's North American domain is that scaling river networks is a non-trivial and
cumbersome task that cannot be fully automated (Arora and Harrison, 2007). In contrast,
conservatively regridding runoff from one spatial resolution to another is a straightforward
process. In addition, it has been shown that routing is not very sensitive to the spatial scale at
which it is performed. Specifically, Arora et al. (2001) evaluated the Arora and Boer (1999) routing
scheme together with the WATROUTE routing scheme at ~350 km and ~25 km spatial resolutions,
respectively, for the Mackenzie River basin. The two routing schemes were driven with the same
runoff. Arora et al. (2001) conclude that for the purpose of realistically modelling streamflow at
the mouth of the rivers in climate models, flow routing at large spatial scales gives similar results
to routing at finer spatial scales. In our study, the difference between the spatial resolution of
runoff ($0.22°$ and $0.44°$) from the CanRCM4 model and routing ($0.5°$) is much smaller than the
Arora et al. (2001) study. As a result, we do not expect that routing at a slightly different spatial
resolution than runoff will lead to significant differences in the simulated streamflow. The routing
scheme needs river flow directions and these are obtained from the Total Integrating Runoff
Pathways (TRIP) data set (http://hydro.iis.u-tokyo.ac.jp/~taikan/TRIPDATA/TRIPDATA.html, last
accessed July 2023) of Oki and Sud (1998). The TRIP data are available at the regular latitude-
longitude grid with the geographic North Pole at its usual location (0° E, 90° N). Figure 1 shows
the river networks at 0.5° resolution based on TRIP data which also identifies the six river basins
investigated in this study. The Fraser River (identified by the light green colour) appears to have
a river mouth over land. This is because the Fraser River drains into the narrow Strait of Georgia
which is not resolved at the 0.5° resolution of the TRIP data set. In addition, the TRIP data set
does not resolve any inland lakes and provides river flow directions over grid cells that are lakes.
This is in fact helpful because it avoids discontinuities in the river network.

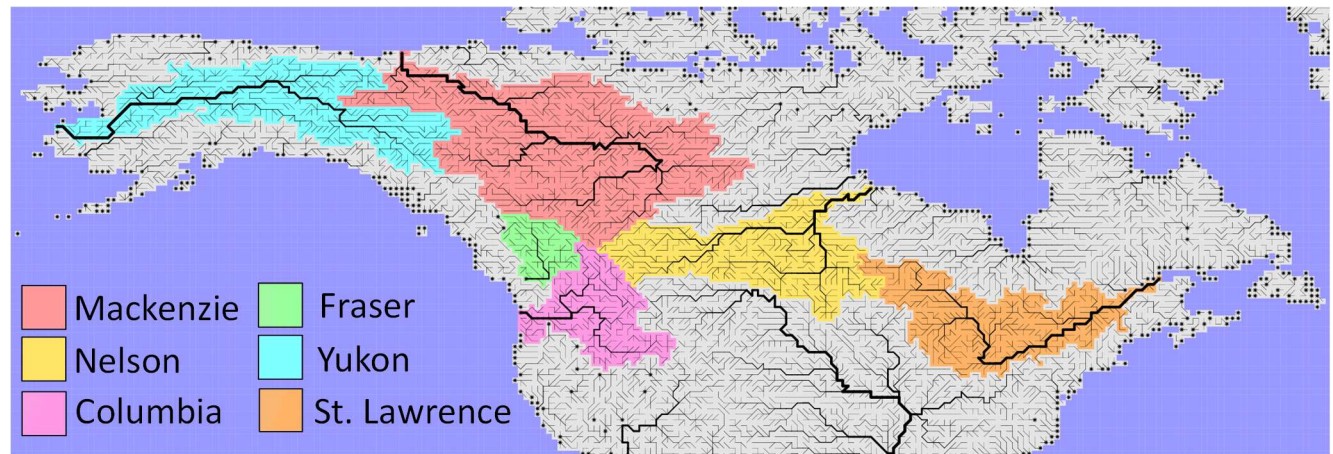

**Figure 1**: River flow networks at 0.5° resolution used in this study. The major river basins for which streamflow and runoff are analyzed in this study are also identified.

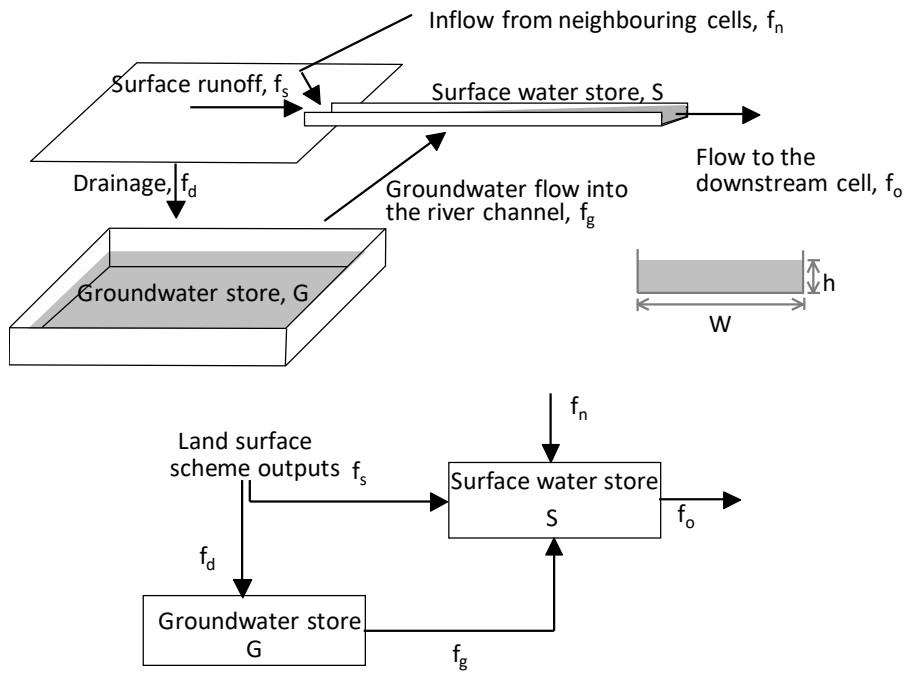

**Figure 2**: Schematic of the Arora and Boer (1999) river routing scheme used in this study to route runoff simulated by CanRCM4.

Figure 2 shows the schematic of the routing scheme which uses surface runoff and drainage outputs from the land surface scheme. The variable velocity routing scheme used here

is described briefly below and more details can be found in Arora and Boer (1999). The water
balance within a grid cell for its surface $S$ (m$^3$) and groundwater $G$ (m$^3$) stores is given by
$$\frac{dS}{dt} = f_s + f_n + f_g - f_o \tag{2}$$

$$\frac{dG}{dt} = f_d - f_g \tag{3}$$

where, $f_s$ and $f_d$ are the surface runoff and drainage (or baseflow) estimates given by the land
surface scheme, $f_n$ and $f_o$ are the surface water inflow from the adjacent upstream neighbouring
grid cell(s) and outflow to the downstream grid cell respectively, and $f_g$ is the groundwater
outflow from the groundwater reservoir to the surface water reservoir within a grid cell as shown
in Figure 2. The fluxes are represented in m$^3$/s.

A river channel is assumed to be rectangular and the width (W) of the river at every point

along the river network is specified a priori. This river width in meters is calculated based on its
geomorphological relationship with mean annual discharge. The surface runoff contributes
directly to the surface water store which is essentially the amount of water in the rectangular
river channel between two grid cells. The flow velocity ($V$, m/s) is calculated using the Mannings
formula (Manning, 1891).
$$V = \frac{1}{r} R^{2/3} s^{1/2} = \frac{1}{r} \left(\frac{A}{P}\right)^{2/3} n^{1/2} = \frac{1}{r} \left(\frac{Wh}{W+2h}\right)^{2/3} n^{1/2} \tag{4}$$

where $r$ is the unitless Mannings roughness coefficient (a default value of 0.04 is used), $A$ is the
area of the river channel (m$^2$), $P$ is the wetted perimeter (m), and $h$ is the depth of water in the
channel (m). The slope $n$ (unitless) of the channel is calculated using elevation difference and the
river length between two grid cells.

The river channel storage S is assumed to be a linear function of outflow discharge, so

$$S = \tau f_o = \frac{L}{V} AV = LA = LWh \tag{5}$$

where $\tau$ is the travel time between the grid cell under consideration and its downstream
neighbor given by $\tau = L/V$, where $L$ is the distance between the grid cells (m). The outflow $f_o$ is
given by

$$f_o = AV = WhV = Wh \frac{1}{r} \left( \frac{Wh}{W+2h} \right)^{2/3} n^{1/2} \tag{6}$$

and substituting (5) and (6) into (2) yields

$$\frac{dh}{dt} = \frac{1}{LW} \left( I - \frac{W^{5/3} h^{5/3}}{r(W+2h)^{2/3}} n^{1/2} \right)^{\square} \tag{7}$$

where $I$ $(m^3/s)$ is the total inflow into a grid cell ($I = f_s + f_n + f_g$ ). Equation (7) describes the flow in
terms of the rate of change of flow depth for a given river section. An explicit forward step finite
difference approximation for (7) yields

$$h(t+1) = h(t) + \frac{\Delta t}{LW} \left( I(t) - \frac{W^{5/3} h(t)^{5/3}}{r(W+2h(t))^{2/3}} n^{1/2} \right)^{\square} \tag{8}$$

Flow velocity and outflow discharge for the river channel at any time step can be obtained using
equations (4) and (6). For the 0.5° resolution used here, a stable solution of (8) is obtained with
$\Delta t$ equal to around 10 minutes. The approach yields dynamically-varying flow depth, velocity,
and discharge through the river channel in response to changing surface and baseflow runoff
inputs from the land surface model.
The groundwater component of the routing model assumes that groundwater storage, $G$,
is a linear function of groundwater outflow, $f_g$.
$$G = \tau_g \, f_g \qquad\qquad (9)$$
The delay in groundwater storage ($\tau_g$) is based on the dominant soil texture type and is set to 10,
35, and 65 days if the dominant soil type in each grid cell is sand, silt, and clay, respectively,
following Arora and Boer (1999). Substituting G in equation (3) yields
$$\tau_g \frac{df_g}{dt} = f_d - f_g \qquad\qquad (10)$$
and following Arora and Boer (1999) we use the following expression
$$f_g(t+1) = f_g(t) \, e^{-\Delta t/\tau_g} + \left(1 - e^{-\frac{\Delta t}{\tau_g}}\right) f_d(t) \qquad\qquad (11)$$
to determine discharge from the groundwater reservoir within a grid cell and to step forward in
time, where a time step $\Delta t$ equal to three hours is used. The simplistic form of equation (11)
allows to use a much larger time step than the time step of 10 minutes required for equation (8).
The routing scheme used here does not consider the flow regulation effect of dams and
reservoirs. It, however, does consider the effect of lakes and ice jams in a simple manner. The
global lake data set from Kourzeneva et al. (2012) is used which prescribes the fractional coverage
of sub-grid lakes and the five Laurentian Great Lakes (Lakes Superior, Michigan, Huron, Ontario,
and Erie). In particular, the flow at the mouth of the St. Lawrence River is affected significantly
by the Great Lakes. The hydraulic residence time of water in the Great Lakes varies from about 2
years for Lake Erie to about 200 years for Lake Superior (Quinn, 1992). As a result, even in the
absence of anthropogenic flow regulation for the St. Lawrence River, we expect the streamflow
at its mouth to show very little seasonality compared to the usual spring peak of Canadian rivers
dominated by snowmelt. The simple approach used here delays the streamflow flowing into a
grid cell with a lake fraction greater than 60% using an e-folding time scale of 300 days similar to
the treatment of the groundwater reservoir (Figure 2) (Arora and Boer, 1999). For the St.
Lawrence River, the effect of delay caused by the Great Lakes is much larger than that of the
anthropogenic flow regulation.
Ice jams and breakups are complex thermal and mechanical events and therefore
challenging to model. They occur on all Canadian rivers with varying degrees and depend on
winter temperatures, the river bathymetry, and the physical and geomorphological conditions of
rivers (Beltaos, 2000; Prowse, 1986). The winter freezing of river water inevitably leads to a slow
down of river flow velocity. When water cannot move downstream, upstream flooding results.
Here, we have used a simple approach that increases Manning's roughness coefficient for the
Mackenzie and the Yukon Rivers (which are the most northerly and therefore affected the most
by ice jams) for the period January to June. The value of Manning's roughness coefficient is
increased linearly from 0.04 to 0.08 from 1 January to 31 January, kept at 0.08 from 1 Feb to 31
May, and then reduced linearly from 0.08 to 0.04 over the period June 1 to 30 June. Chen and
She (2020) report the trend in river ice breakup dates for the Mackenzie and Yukon Rivers to be
around -0.3 and -1.3 days/decade for the 1950-2016 period, where the negative sign indicates
that the ice breakup is occurring earlier. Assuming the same trend, the breakup dates would
occur about 2.5 and 11 days earlier towards the end of this century, respectively, for the
Mackenzie and Yukon rivers. This simple approach reduces the river flow velocity during the
months that are most affected by river ice jams. Although this is not a perfect nor a complete
approach this simple treatment allows to improve the streamflow seasonality for the Mackenzie
and Yukon rivers. For the southerly Fraser and Columbia rivers such treatment was not necessary.
Consideration of a higher roughness coefficient for the St. Lawrence River to account for ice jams
does not affect its streamflow's seasonality (or rather the lack of it) which is overwhelmingly
determined by the delay and storage caused by the Great Lakes.
**2.3 Modelled and observation-based data**
The CMIP5 historical simulation covers the period 1850-2005 and the future scenarios
cover the period 2006-2100. We used daily runoff from CanRCM4 from its $0.22°$ North American
domain for the 20 years 1986-2005 from one ensemble member of the historical simulation and
for the 20 years 2081-2100 from one ensemble member each for the two future scenarios (RCP
4.5 and RCP 8.5, Moss et al. (2010)). The RCP 8.5 is the highest baseline emissions scenario where
future development is based on continuous fossil-fuel development. As a result, $CO_2$ emissions
and concentrations increase throughout the $21^{st}$ century and $CO_2$ concentration in the year 2100
is around 1100 ppm. RCP 4.5 is a moderate emissions scenario in which emissions peak around
2040 and then decline: as a result $CO_2$ somewhat stabilizes to around 550 ppm by the year 2100.
Since the CanRCM4 data are available on a rotated latitude-longitude grid and the river routing
is performed on a regular latitude-longitude grid (following the TRIP data), the runoff data from
CanRCM4 are conservatively regridded to the global $0.5°$ grid using climate data operators (CDO)
(https://code.mpimet.mpg.de/projects/cdo/embedded/index.html#x1-7170002.12.5,          last
accessed Dec 2023) as mentioned earlier. These runoff data are then used as input into the
routing model. The 20-year runoff data (1986-2005 for the historical simulation, and 2081-2100
for the future scenarios) are concatenated into a 40-year time series for each simulation
(historical, RCP 4.5, and RCP 8.5). These data are then input into the routing model and the last
20 years of simulated streamflow are analyzed. The 20-year spin-up is sufficient to allow the
surface and groundwater stores to fill up and reach equilibrium. The simulated precipitation and
temperature from CanRCM4 are compared against observation-based data from the CRU TS 4.07
product (Harris et al., 2020).

The simulated streamflow is compared against observation-based estimates obtained

from the Global Runoff Data Centre (GRDC) for the stations that are closest to the river mouths.
Table 1 lists the drainage areas of all rivers considered in this study as discretized in the TRIP data
set and at the stations closest to the river mouth. For the Columbia River, which is heavily
regulated, we obtain an estimate of the naturalized flow with no regulation and no irrigation
provided by the Bonville Power Administration (BPA) for the station VAN (near Vancouver,
Washington, USA) (https://www.bpa.gov/energy-and-services/power/historical-streamflow-
data;https://www.bpa.gov/-/media/Aep/power/historical-streamflow-reports/historic-
streamflow-nrni-flows-1929-2008-corrected-04-2017.csv, last accessed July 2023). The drainage
area of the Columbia River upstream of the VAN station is 616960 $km^2$ and does not include
discharge contributions from three tributaries (Willamette, Cowlitz, and Lewis Rivers). Of these
three tributaries, the contribution from Willamette is the largest. We obtained naturalized
streamflow for the Willamette River at the station SVN (drainage area 25,600 $km^2$) also from
BPA's website (https://www.bpa.gov/-/media/Aep/power/historical-streamflow-
reports/correction-20220801.zip, from the file SVN6ARF_daily_COR.xlsx) and added it to the
naturalized streamflow at the station VAN. This yields naturalized streamflow for the entire
Columbia River basin, except the smaller Cowlitz, and Lewis Rivers, and represents a drainage
area of 642,560 km$^2$ (see Table 1).
The Nelson River is affected by two large lakes, Lake Winnipeg and Lake Manitoba, and it
is also heavily regulated. It currently has five dams towards the end of its journey as it flows into
Hudson Bay. There are no upstream gauging stations close to the first upstream dam. In addition,
water is also diverted from Churchill to the Nelson River. We were unable to obtain naturalized
flow for the Nelson River from the Manitoba hydroelectricity company. Due to anthropogenic
flow regulation on the Nelson River, the present-day streamflow shows very little seasonality (as
shown later). As a result, we do not evaluate the simulated daily or monthly streamflow for the
Nelson River and focus only on its mean annual value.
**Table 1**: Comparison of river basin areas as represented in the TRIP data and at the gauging
station closest to the river mouth for the river basins considered in this study as obtained from
the GRDC.

| River basin | River basin area (million km$^2$) | | Gauging station |
| | in the TRIP data set | at the gauging station closest to the river mouth | |
|---|---|---|---|
| Mackenzie | 1.74 | 1.66 | Arctic red river |
| Yukon | 0.85 | 0.83 | Pilot Station |
| Columbia | 0.66 | 0.64 | See section 2.3 |
| Fraser | 0.23 | 0.22 | Hope |
| Nelson | 1.07 | 1.06 | Long Spruce generating station |
| St. Lawrence | 1.11 | 0.77 | Cornwall, Ontario |



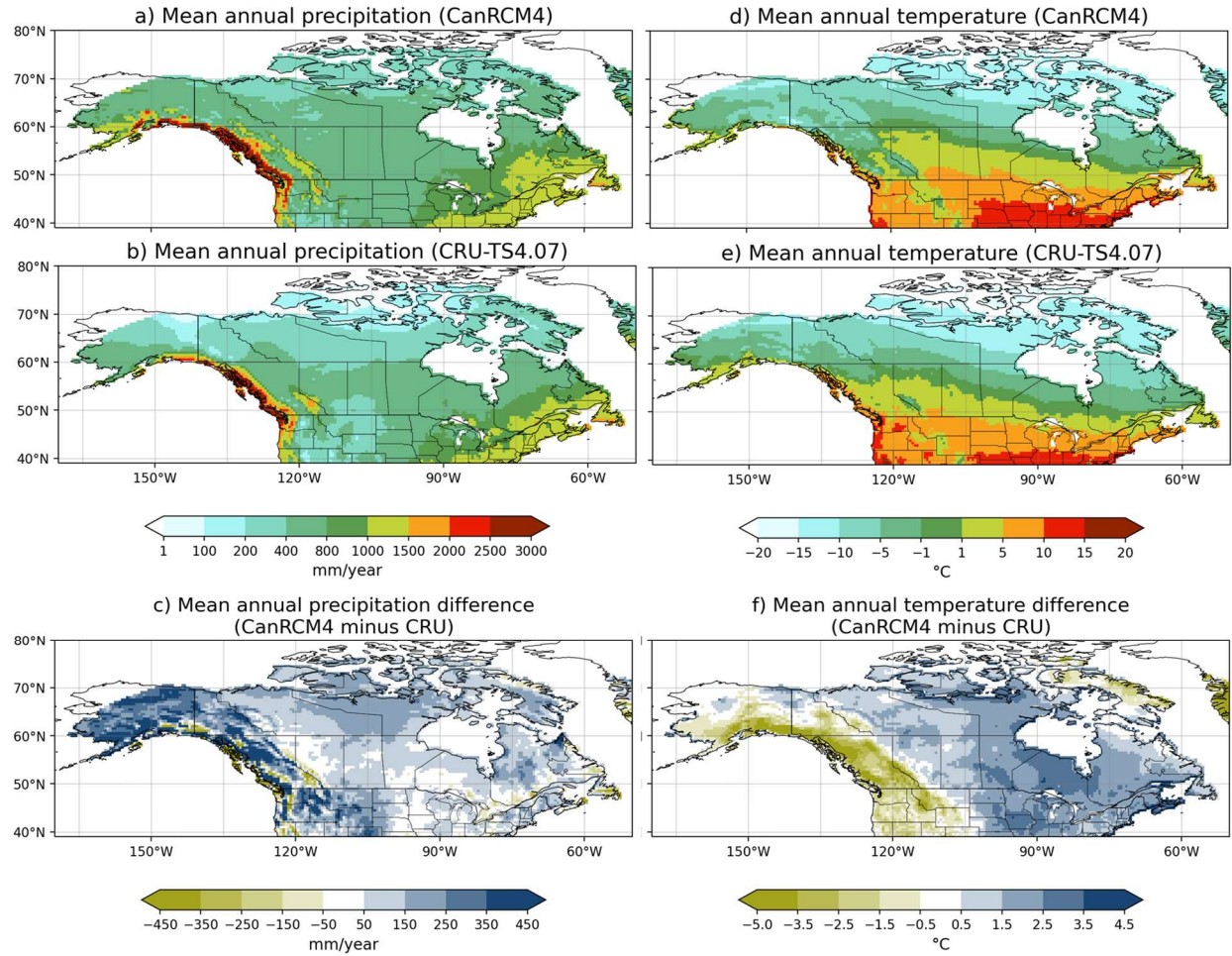


**Figure 3**: Comparison of CanRCM4 simulated precipitation (left column) and temperature (right
column) with observation-based estimates from the CRU TS 4.07 data set for the period 1986-

387 2005.


**3. Results**
**3.1 Present-day precipitation, temperature, and streamflow**
Figure 3 compares the geographical distribution of mean annual precipitation (left
column) and temperature (right column) simulated by CanRCM4 to observation-based estimates
from the CRU TS 4.07 data set (referred to as CRU from here on) for the 1986-2005 period.
Although the six river basins considered in this study do not cover the entire Canadian region, for
completeness the plots are shown for the whole of Canada and south up to 39 °N to include the
southern edge of the Columbia River basin. In Figure 3, while CanRCM4 broadly simulates the
geographical distribution of temperature and precipitation reasonably realistically, there are
differences compared to the CRU data set. CanRCM4 generally simulates higher precipitation
over Canada and more so to the west of the Rockies (Figure 3c) compared to observations. The
model simulates cooler than observed temperatures to the west of the Rockies and higher than
observed temperatures to the east of the Rockies (Figure 3f).  This is likely related to the
representation of topography in the model. The overall somewhat higher precipitation in
CanRCM4 over North America is also noted by Alaya et al. (2019) who compared probable
maximum precipitation (PMP) calculated using CanRCM4 data to estimates based on several
reanalyses. Alaya et al. (2019) concluded that among the three reanalyses they considered,
CanRCM4 compared best with the National Centre for Environmental Prediction's (NCEP) Climate
Forecast System Reanalysis.
Figure 4 compares the simulated annual cycle of temperature (left column) and
precipitation (middle column) over the six river basins (Figure 1) selected in this study with
observation-based estimates from CRU. The right-hand side column compares simulated
streamflow for the six river basins with observation-based estimates from the GRDC.  The basin-
averaged values of temperature and precipitation are calculated by area weighting the values in
the individual grid cells that lie inside a given river basin according to the TRIP data (Figure 1).
The plots also show the mean annual values (dashed lines) on the plot and their magnitude in
the legend. Figure 4 shows that overall CanRCM4 simulated basin-wide averaged temperatures
compare reasonably well with observation-based estimates based on the CRU data for the
Mackenzie and the Yukon River basins. For the Columbia and Fraser, the simulated temperatures
are lower for most months, and for the Nelson River basin, the CanRCM4 simulated temperatures
are higher compared to the CRU data. The seasonal cycle of temperature compares well with the
observation-based estimates from CRU data. Compared to temperature, there are larger
differences in simulated CanRCM4 precipitation compared to the CRU data. Although CanRCM4
simulates the seasonality of precipitation reasonably well compared to the CRU data, simulated
precipitation is higher for all river basins, consistent with Figure 3c. The comparison with the CRU
data provides useful insights into simulated quantities. Specifically, despite the difference in the
magnitudes, CanRCM4 provides a reasonable representation of the seasonality of precipitation,
for example, higher winter precipitation in the southern Fraser and Columbia basins, and higher
summer precipitation in the northern Mackenzie and Yukon basins. However, all observation-
based data sets (including CRU) have their limitations. Wong et al. (2017) compared several
gridded observation-based precipitation data sets over Canada and found that they all have
limitations and the data sets compared best with gauge-based precipitation data in summer,
followed by autumn, spring, and winter in order of decreasing quality. Sun et al. (2018) compare
global precipitation from 22 gauge-, satellite-, and reanalysis-based products, including CRU, and
quantify the uncertainty in the different precipitation estimates over timescales ranging from
daily to annual.  Shi et al. (2017) evaluated the CRU precipitation over large regions of China and
found that CRU underestimates precipitation in that region compared to rain gauge records.
Furthermore, observation-based precipitation datasets also generally tend to underrepresent
total precipitation in mountainous western Canada (where the Yukon, Mackenzie, Fraser, and
Columbia River basins are located) due to low station density at high elevations (Werner et al.,
2019). In the end, the objective of the comparison of the simulated climate with CRU
observations is to evaluate if the model climate is reasonably realistic for the present day. The
assumption behind using direct output from climate models is that despite the biases in the
simulated current climate it is possible to deduce meaningful information about the effect of
climate change using the change in simulated quantities.

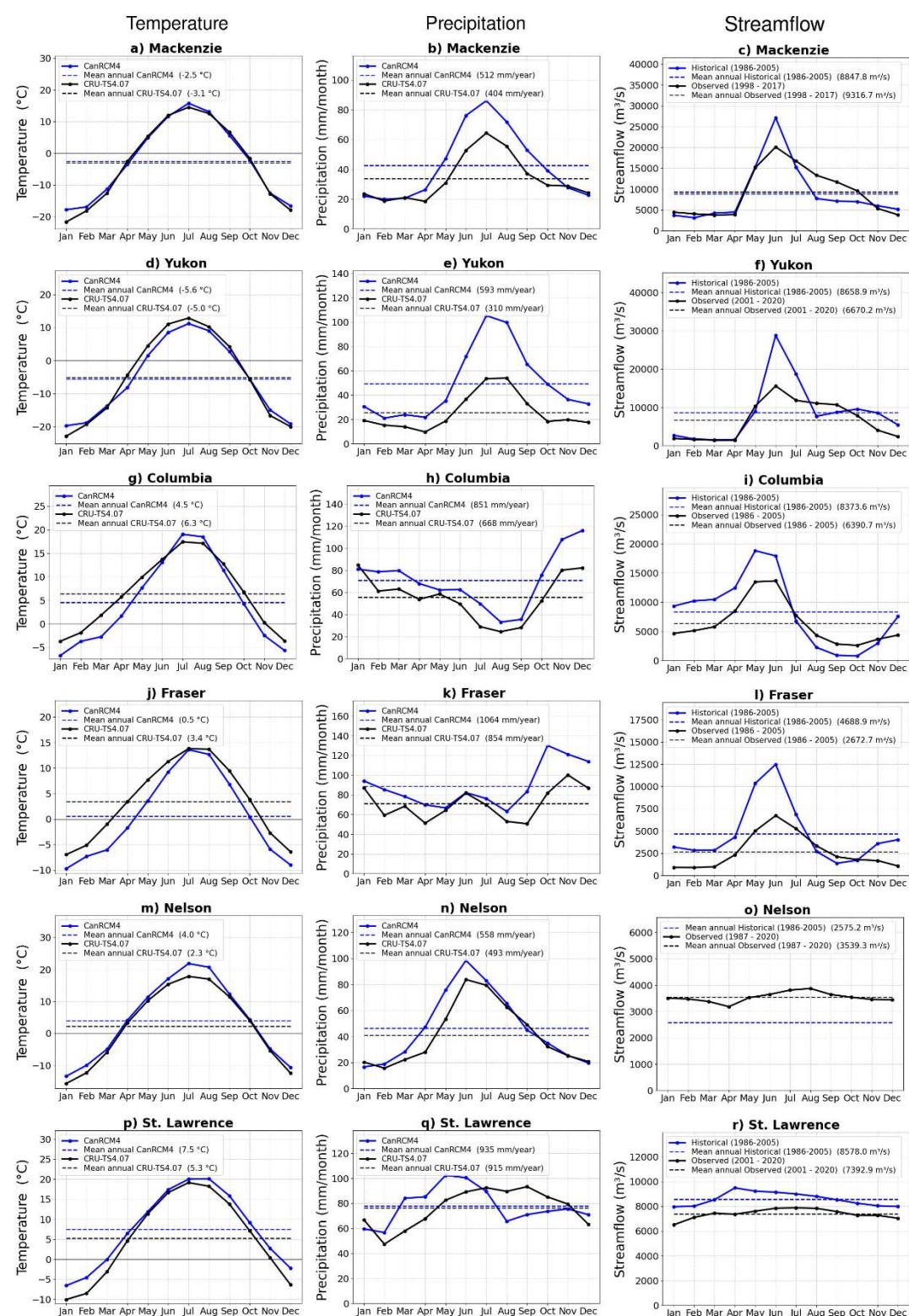


**Figure 4**: Comparison of the annual cycle of basin-wide averaged CanRCM4 simulated
temperature (left column) and precipitation (middle column) with observation-based estimates
from the CRU TS 4.07 data set for the period 1986-2005. The right-hand side column compares
simulated streamflow with observations from the GRDC. In the absence of the consideration of
anthropogenic flow regulation for the Nelson River only its simulated mean annual streamflow
value is evaluated.

The differences in simulated climate between CanRCM4 and the observation-based
climate in CRU for the present day affect simulated streamflow as expected. The simulated mean
annual streamflow is higher for four out of six river basins considered (Yukon, Columbia, Fraser,
and St. Lawrence) primarily because of the higher simulated precipitation. Simulated
precipitation is also higher for the Mackenzie River basin, but the mean annual simulated
streamflow compares well with its observation-based estimate. Possible reasons for reasonably
realistic annual simulated streamflow despite higher precipitation could be biases in the CRU
data set itself (e.g., underrepresentation of total annual precipitation), or higher simulated
evaporation in CanRCM4 (although simulated summer temperatures compare well with the CRU
data). Finally, the simulated mean annual streamflow for the Nelson River is lower than its
observation-based estimate despite somewhat higher simulated precipitation than the CRU data.
The most likely reason for this is the diversion from the Churchill River into the Nelson River which
started in 1976 to increase the water flow to larger generating stations on the lower Nelson River.
The Manitoba government estimates that an average of 25% more water flows into the lower
Nelson       River       due       to       the       Churchill       River       Diversion       (CRD)
(https://www.gov.mb.ca/sd/water/water-power/churchill/index.html, last accessed Sep. 2023).
The seasonality of streamflow for the Mackenzie, Yukon, and Fraser Rivers is dominated by the
spring snowmelt with the peak occurring in June for both simulated and observed streamflow.
The simulated streamflow for the Columbia and Fraser rivers peaks at the right time but there is
more simulated streamflow during the winter months when precipitation is also higher than
observed. For the Mackenzie and Yukon rivers although the mean annual simulated and observed
streamflow are comparable their seasonal distribution is not. The simulated streamflow peak for
these rivers is higher due to the simple treatment of ice jams which is not sufficient to hold the
water in the river channel and then release it slowly as ice jams slowly dissipate in the spring and
summer months, as the observed streamflow indicates. Finally, for the St. Lawrence River, there
is little seasonality in observed streamflow due to the delay caused by the Great Lakes and
anthropogenic flow regulation. The lack of strong seasonality simulated in simulated streamflow
for the St. Lawrence River is caused entirely due to the delay caused by the Great Lakes (section

2.2).

Overall the spatial distribution of precipitation and temperature over Canada (Figure 3),

and the seasonality of these two primary climate drivers for the river basins considered in this
study (Figure 4), compare reasonably well with observation-based estimates from the CRU data,
although there are differences in the absolute magnitude of these variables. The resulting
seasonality of streamflow has limitations due to three factors: 1) the biases in the driving climate
from CanRCM4, 2) the biases in the land surface component of CanRCM4 which partitions
precipitation into evaporation and runoff, 3) the lack of calibration of the land surface component
to specific river basins, and 4) the lack of processes in the routing component including the
limitation of not being able to treat ice jams comprehensively. Despite these limitations, the
simulated streamflow captures the broad seasonal patterns with higher values during the spring
snow melt and lower values during the winter months as observations show.

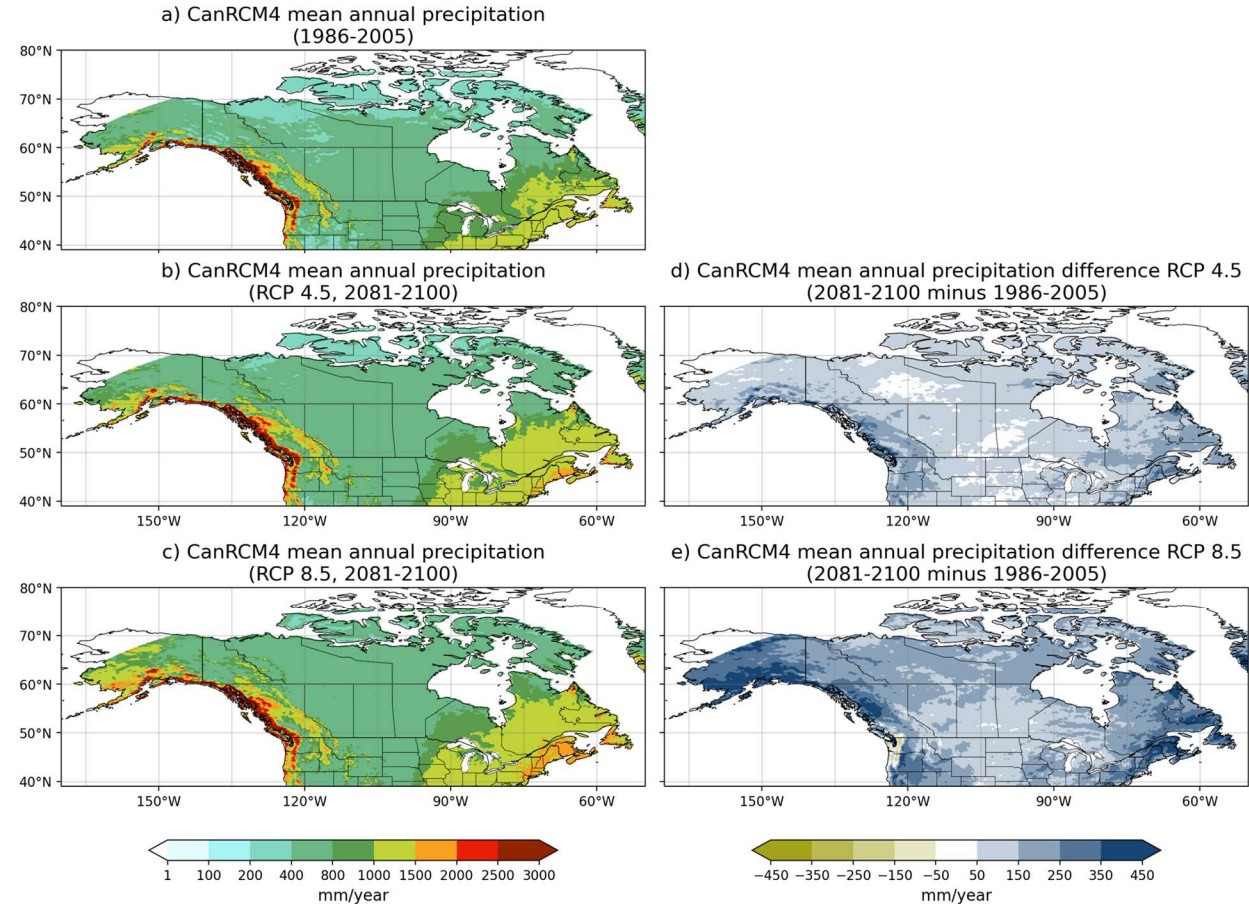

**Figure 5**: Comparison of CanRCM4 simulated precipitation for the 1986-2005 and for the 2081-2100 periods for RCP 4.5 and 8.5 scenarios.

**3.2 Changes in future climate and streamflow**

Figures 5, 6, and 7 show the changes in CanRCM4 simulated precipitation, temperature, and runoff for the period 2081-2100, for both RCP 4.5 and 8.5 scenarios, compared to the 1986-2005 period from the historical simulation. Over Canada, simulated precipitation and temperature increase almost everywhere and in both scenarios. As expected, the magnitude of precipitation and temperature change is higher for the RCP 8.5 than the RCP 4.5 scenario. Simulated precipitation increases are higher in the coastal western and eastern Canadian regions than in central and northern parts of Canada. The central Canadian region sees the lowest

increase in precipitation in both scenarios. Simulated temperature increases, as expected, are
higher at higher latitudes due to polar amplification of the temperature change associated with
the snow- and ice-albedo feedbacks. In the RCP 4.5 and 8.5 scenarios, the simulated temperature
changes vary from about 3 °C and 6 °C, respectively, in the south, to about 6 °C and 11 °C, in the
north. The parent climate model (CanESM2) on which CanRCM4 is based has an equilibrium
climate sensitivity of 3.7 °C, somewhat on the higher side, compared to the range of 1.5 °C to 4.5
°C amongst climate models that contributed to CMIP5 (Schlund et al., 2020). As a result, we also
expect the magnitude of simulated changes to be somewhat higher than a model with average
climate sensitivity.








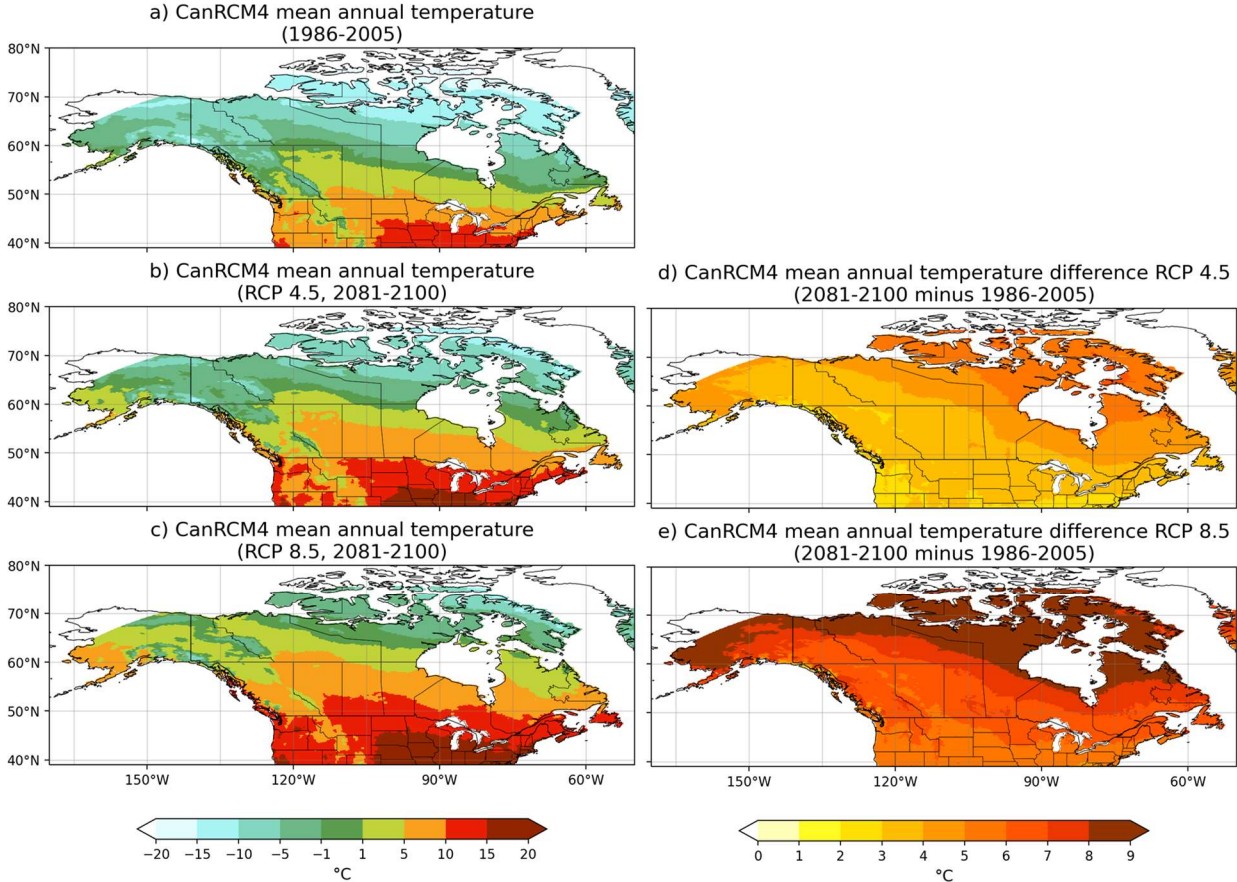

**Figure 6**: Comparison of CanRCM4 simulated temperature for the 1986-2005 period and for the
2081-2100 periods, for RCP 4.5 and 8.5 scenarios.

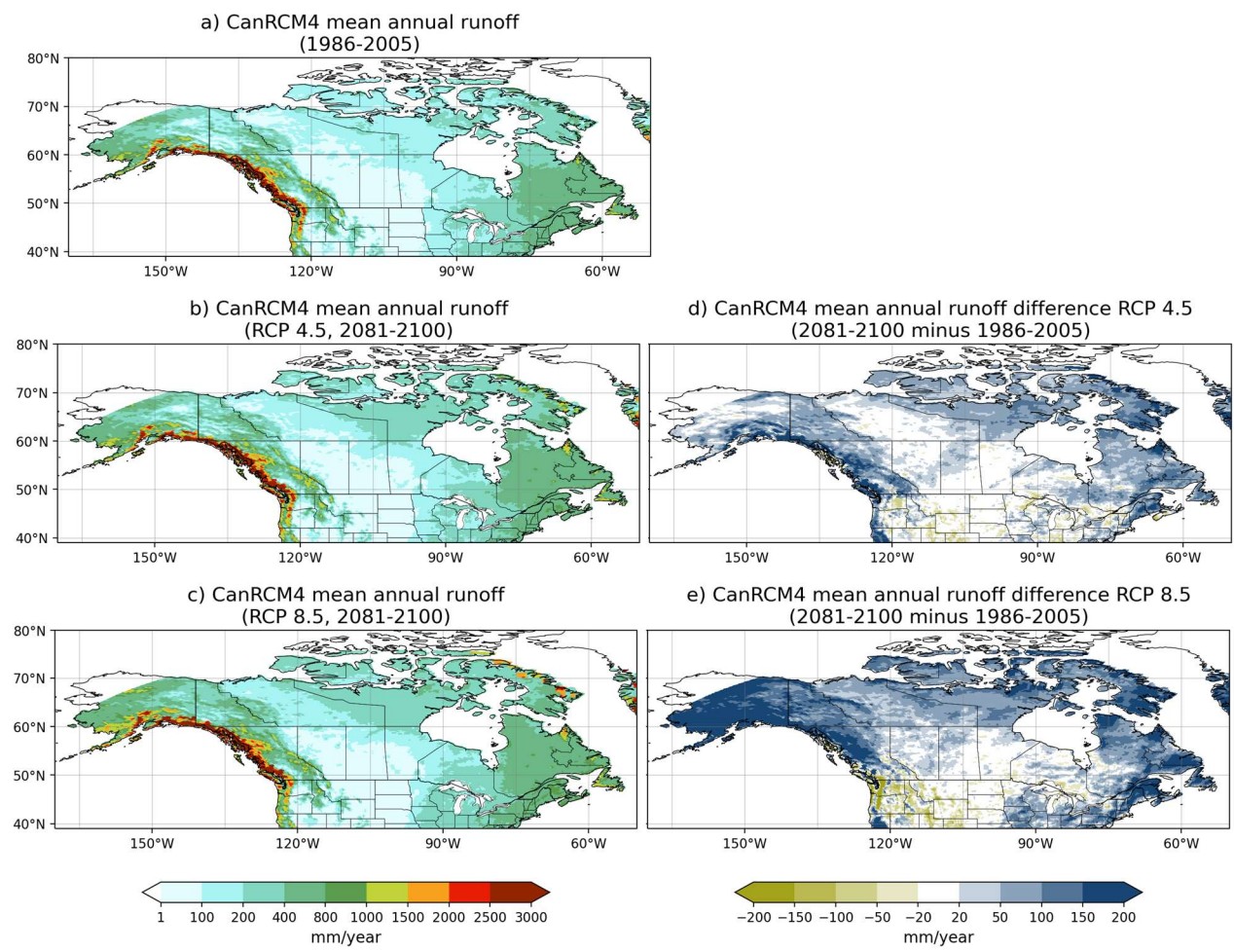


**Figure 7**: Comparison of CanRCM4 simulated runoff for the 1986-2005 period and the 2081-2100 periods, for RCP 4.5 and 8.5 scenarios.


In Figure 7 runoff increases generally everywhere in Canada for the RCP 4.5 and RCP 8.5 scenarios
with larger changes on the west and east coasts, and in northern Canada, following a similar
pattern of changes in precipitation. Runoff reduces in parts of the southern Columbia River basin
in the United States in the RCP 4.5 scenario, and these decreases become more pronounced and
widespread over the north-western Pacific region in the RCP 8.5 scenario including the Fraser
River basin in Canada.

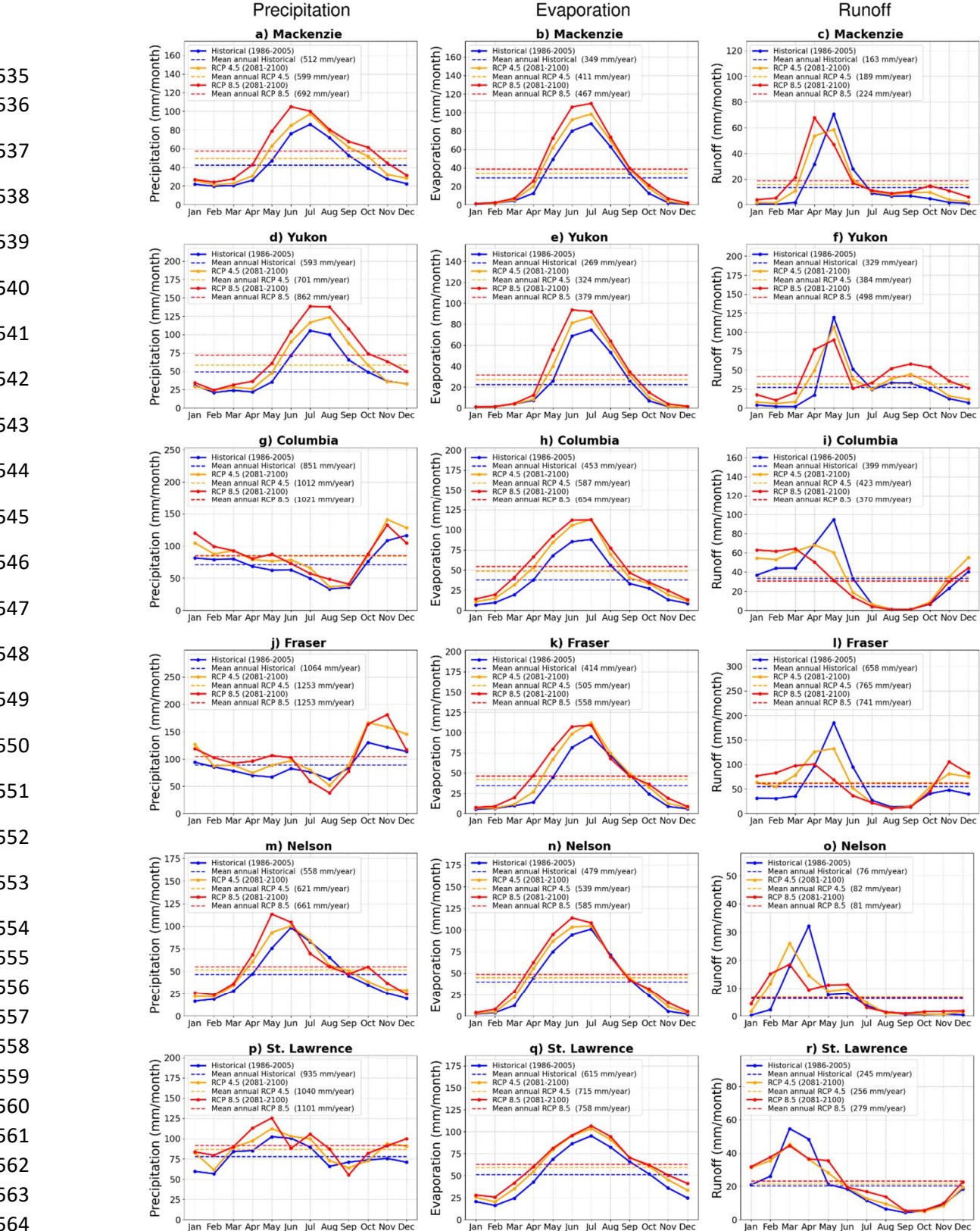

**Figure 8**: Comparison of the annual cycle of basin-wide averaged CanRCM4 simulated water budget components for each river basin for the historical (1986-2005) period and the two future scenarios RCP 4.5 and 8.5 (2081-2100): precipitation (left column), evaporation (middle column), and runoff (right column).

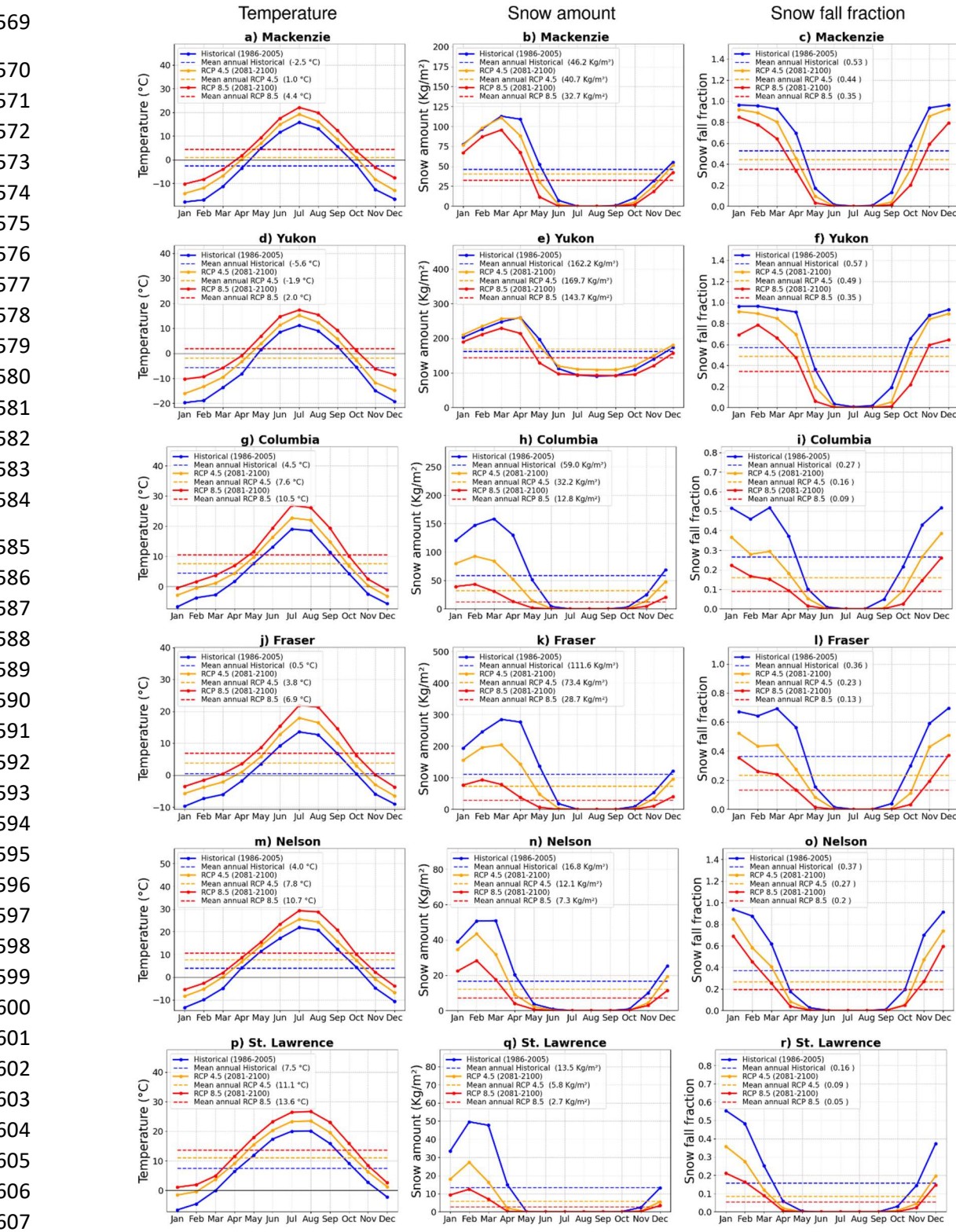

**Figure 9**: Comparison of the annual cycle of basin-wide averaged CanRCM4 simulated temperature (left column), snow water equivalent amount (middle column), and snowfall fraction (right column) for the historical (1986-2005) period and the two future scenarios RCP 4.5 and 8.5 (2081-2100).

Figure 8 shows the annual cycle of the simulated water budget components
(precipitation, evaporation, and runoff) for the six river basins considered in this study for the
historical (1986-2005) period and the two future scenarios, RCP 4.5 and 8.5 (2081-2100). As in
Figure 4, the mean annual values are shown as dashed lines, and their magnitude is noted in the
legend.
**Table 2**: Evaporation and runoff ratios for the six river basins simulated by CanRCM4 for the
historical period (1986-2005) and the two future scenarios (RCP 4.5 and 8.5, 2081-2100). The
evaporation (runoff) ratio is the ratio of mean annual evaporation (runoff) to precipitation.

| River basin | Evaporation ratio (E/P) | | | Runoff ratio (R/P) | | |
|---|---|---|---|---|---|---|
| | Historical (1986-2005) | RCP 4.5 (2081-2100) | RCP 8.5 (2081-2100) | Historical (1986-2005) | RCP 4.5 (2081-2100) | RCP 8.5 (2081-2100) |
| Mackenzie | 0.682 | 0.686 | 0.675 | 0.318 | 0.316 | 0.324 |
| Yukon | 0.454 | 0.462 | 0.440 | 0.555 | 0.548 | 0.579 |
| Columbia | 0.532 | 0.580 | 0.641 | 0.469 | 0.418 | 0.362 |
| Fraser | 0.389 | 0.403 | 0.445 | 0.618 | 0.611 | 0.591 |
| Nelson | 0.858 | 0.868 | 0.885 | 0.136 | 0.132 | 0.123 |
| St. Lawrence | 0.664 | 0.686 | 0.684 | 0.314 | 0.294 | 0.302 |


The evaporation (E/P) and runoff (R/P) ratios for the six river basins for the historical period and
the two future scenarios are shown in Table 2 and allow to see how the partitioning of
precipitation into evaporation and runoff changes with climate. For the mean annual values of P,
E, and R reported in Figure 8, P is balanced to within 1% by E+R for all river basins (except St.
Lawrence) and all scenarios, except for the Yukon (for RCP 8.5) and Fraser River basins (for RCP
4.5 and 8.5) for which (E+R) is higher than P indicating that $\Delta S$ is not zero (see equation 1). As a
result, (E/P) and (R/P) also add to one for all river basins except for the Yukon (RCP
8.5,$(E + R)/P = 1.02$) and the Fraser River (RCP 4.5, $(E + R)/P = 1.014$, and RCP 8.5,
$(E + R)/P = 1.036$) basins. For the St. Lawrence River basin, the imbalance is around 2%
because of the presence of the Great Lakes which had to be excluded from the river basin mask.
Since basin-wide averaged calculations are done at 0.5° latitude-longitude resolution, and the
actual domain of CanRCM4 is on a rotated latitude-longitude projection this led to slightly more
rounding errors for the St. Lawrence than other river basins.

For all river basins considered, precipitation increases for both future scenarios with the

increase being larger for the RCP 8.5 scenario consistent with Figures 5d and 5e. The response of
evaporation to changes in climate is expected. The increase in precipitation and temperature
yields an increase in evaporation for future scenarios for all river basins. Simulated runoff does
not increase as much as precipitation since evaporation also increases. The runoff ratio, in Table
2, increases for the northerly Mackenzie and the Yukon River basins while it decreases for the
southerly Nelson, St. Lawrence, and especially for the Fraser and Columbia River basins which
are characterized by milder climate owing to their location in the Pacific north-western region.
This is because the increase in precipitation is more than enough to compensate for the increase
in evaporation (associated with a warmer climate) for the northern river basins but not for the
southern ones (as seen earlier in Figure 7 where runoff begins to decrease in parts of the
Columbia and Fraser River basins). The absolute runoff amount in Figure 8 increases for the
Mackenzie and Yukon River basins, in the RCP 4.5 and 8.5 scenarios compared to the historical
simulation, but doesn't change much for the Columbia, Fraser, Nelson, and St. Lawrence River
basins. However, the seasonality of runoff changes for all river basins, and the peak in simulated
runoff either occurs earlier in the year, occurs with reduced magnitude, or both. Canadian rivers
are dominated by spring snowmelt and this runoff behaviour is associated with snow melt
occurring earlier in the year in the RCP 4.5 scenario than in the historical simulation and occurring
even earlier in the RCP 8.5. This is seen in Figure 9 which shows the simulated annual cycle of
temperature changes, snow amount, and snowfall as a fraction of total precipitation for the
historical period and the two RCP scenarios for the six river basins. In Figure 9 the mean annual
temperature increases from the historical period to the RCP 4.5 scenario, and from the RCP 4.5
to RCP 8.5 scenario, are between 3 and 3.5 °C for the six river basins considered here. The middle
column of Figure 9 shows that in addition to earlier snowmelt the amount of snow in the winter
months decreases for all river basins with climate warming. The only exception to this is the
Yukon River basin in which the mean annual snow amount increases marginally in the RCP 4.5
scenario (Figure 9e). As expected, the fraction of precipitation falling as snow also decreases with
climate warming for all river basins (right column, Figure 9).

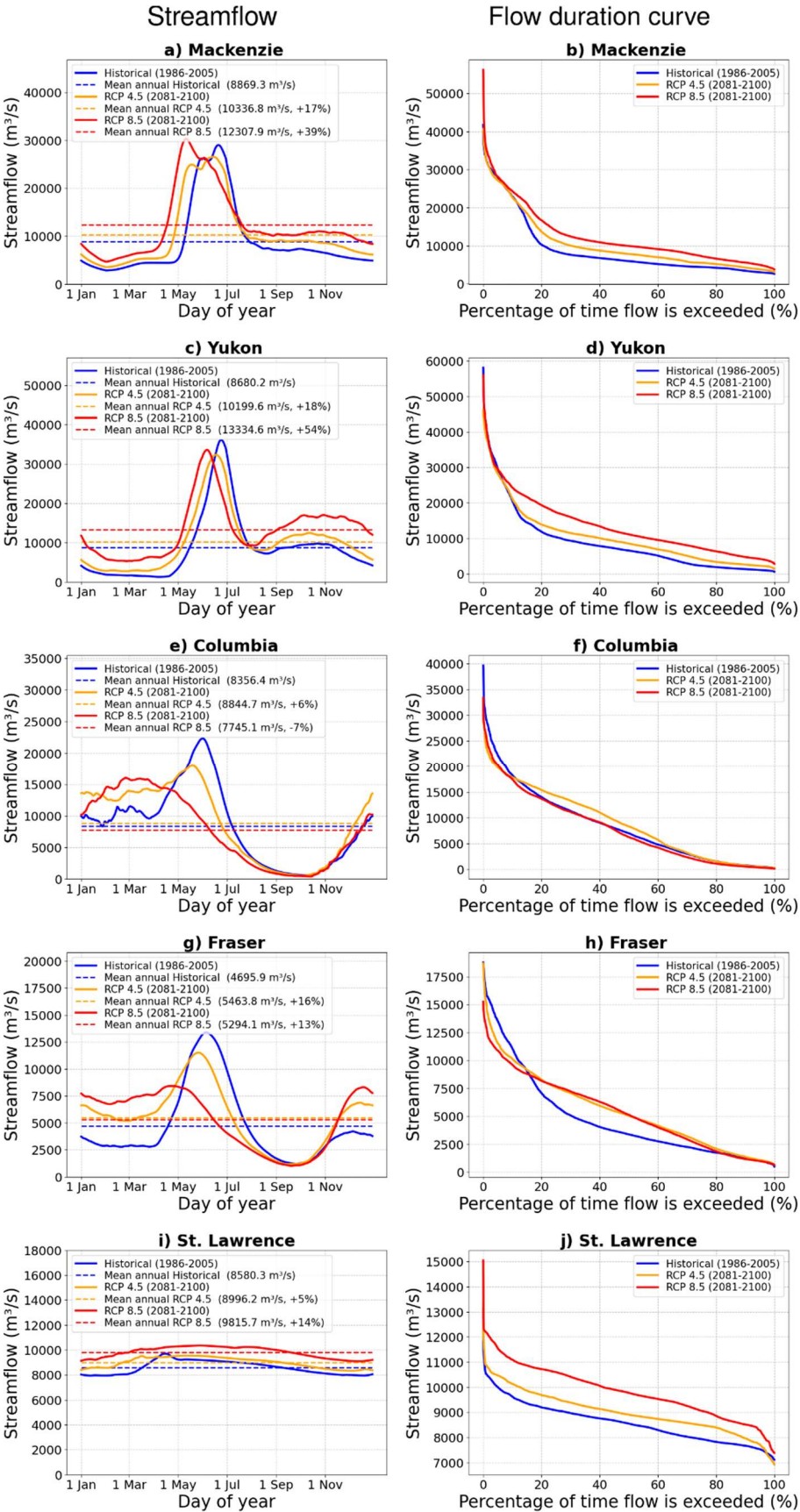

**Figure 10**: Comparison of the simulated daily streamflow (left column) and flow duration curves (right column) for the historical (1986-2005) period and the two future scenarios RCP 4.5 and 8.5 (2081-2100) for the river basins considered. The Nelson River is excluded for which we only evaluated annual streamflow values that are mentioned in the text.

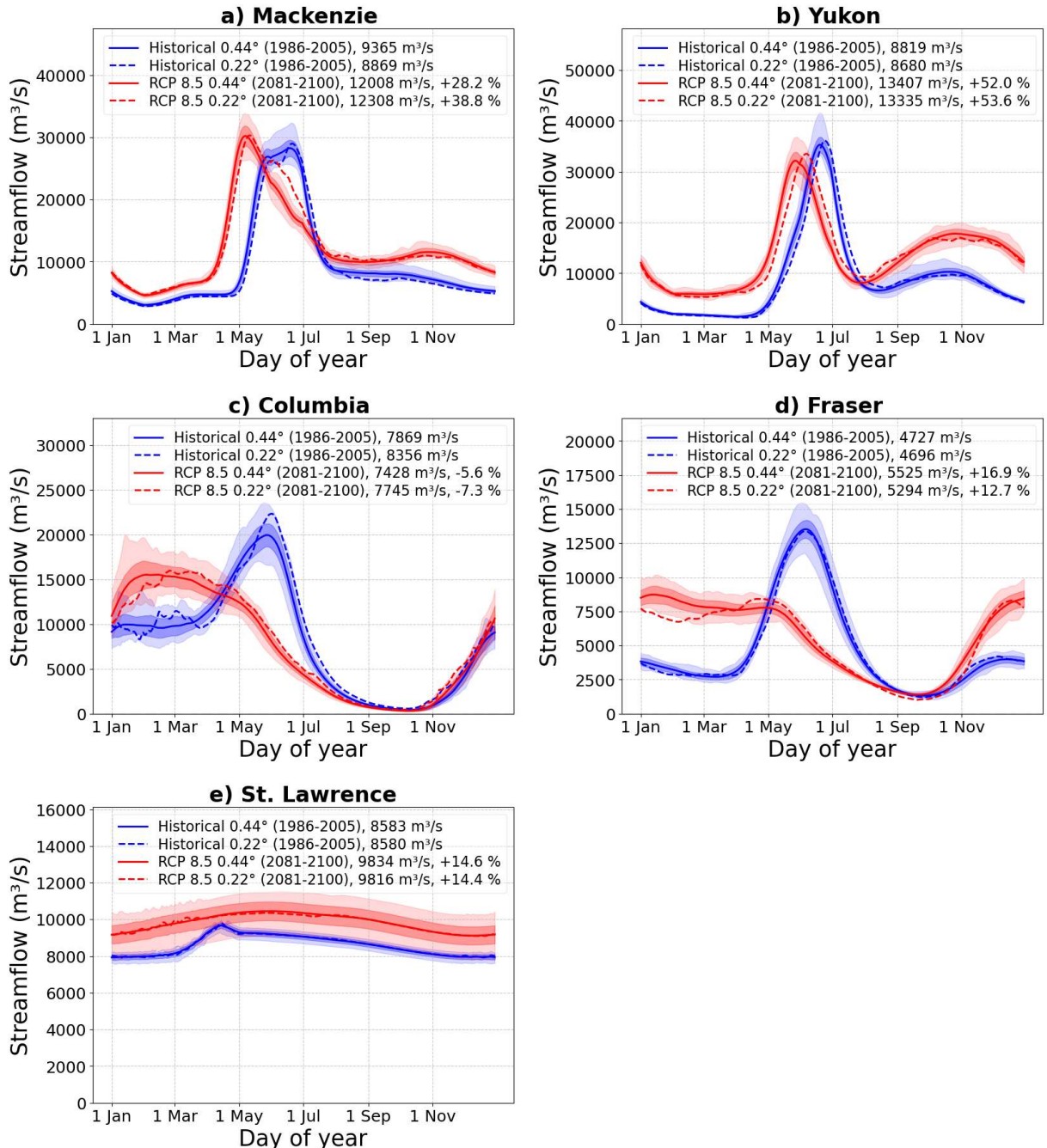

**Figure 11**: Comparison of the simulated daily streamflow for the historical (1986-2005) period and the RCP 8.5 scenario (2081-2100) for the river basins considered in this study from the 0.22° and 0.44° simulations. The results from the 0.22° simulations (shown earlier in Figure 10) are shown as dashed lines. The uncertainty range for the 0.44° simulations is based on results from CanRCM4's 50-member large ensemble. The solid lines indicate the mean across 50 members the light shading indicates the full range, and the dark shading indicates the mean ± one standard deviation range, for the 0.44° simulations. The Nelson River is excluded for which

only annual streamflow values are analyzed.

Figure 10 compares simulated daily streamflow and flow duration curves averaged over
the historical (1986-2005) period with those averaged over the two future scenarios RCP 4.5 and
8.5 (2081-2100) for the river basins considered here excluding the Nelson River. The flow
duration curves are calculated using daily streamflow values. The legends in Figure 10 for the
streamflow figures in the left column show mean annual values but also the change from the
simulated historical values for the RCP 4.5 and 8.5 scenarios. The mean annual streamflow
increases for all rivers for both the RCP 4.5 and 8.5 scenarios, except for the Columbia River for
the RCP 8.5 scenario (-7%). The increase in simulated annual streamflow is largest for the
Mackenzie (+16%, +39%) and Yukon Rivers (+17%, +53%) for the RCP 4.5 and 8.5 scenarios, due
to higher precipitation increase in these two basins (Figure 8). The increase in annual streamflow
for other rivers is smaller and between 6% and 14%. Daily streamflow and flow duration curves
are not shown for the Nelson River because we do not consider anthropogenic flow regulation,
as mentioned earlier. The simulated mean annual streamflow for the Nelson River increases from
2556.6 m$^3$/s (for the 1986-2005 period) to 2774.8 and 2723.8 m$^3$/s for the RCP 4.5 (+9%) and 8.5
(+7%) scenarios, respectively (for the period 2081-2100).
The changes in streamflow seasonality are larger for the southerly Columbia and Fraser
Rivers than for the northerly Mackenzie and Yukon Rivers. The peak daily streamflow for the
Yukon River still occurs in June given it's the coldest river basin (Figure 4d) and the streamflow
seasonality is still dominated by the spring snowmelt. The simulated daily peak streamflow for
the Yukon River occurs on 24 June for the historical period (1986-2005), and 18 June and 6 June,
respectively, for RCP 4.5 and 8.5 scenarios for the period 2081-2100. Streamflow for the Yukon
River also begins to increase earlier due to earlier snowmelt (Figure 9e). While the spring peak
streamflow reduces in both RCP 4.5 and 8.5 scenarios during June and part of July, streamflow
increases for most other months for the Yukon River. The Mackenzie River shows similar
behaviour to the Yukon River in terms of earlier shifts of spring streamflow peaks with climate
warming but the spring peak is higher for the RCP 8.5 scenario.  The mean simulated daily peak
streamflow for the Mackenzie River occurs on 21 June for the historical period (1986-2005), and
14 June and 11 May, respectively, for RCP 4.5 and 8.5 scenarios for the period 2081-2100. Similar
to Yukon, although the streamflow is lower for the Mackenzie River during June and part of July,
it increases for most other months. The corresponding changes in streamflow are also seen in
the flow duration curves. For these two rivers the frequency of the occurrence of flows that occur
greater than about 5% of the time in the historical simulation increases in the future. The
Columbia and the Fraser Rivers experience much larger changes in their seasonality as their
primarily snow-dominated flow regimes change to more hybrid flow regimes. The snowmelt-
driven streamflow peak in spring is reduced considerably for future scenarios since a lower
fraction of fall, winter, and spring precipitation falls as snow. As a result, streamflow increases
from October to April since precipitation falling as rain, as opposed to snow, yields runoff that
runs straight into the rivers. Additionally, the large reduction in snowpack volume together with
earlier melt (Figure 9k and 9h) affects the seasonality of the Fraser and Columbia Rivers
streamflow and causes pronounced shifts in peak flows. The mean simulated daily peak
streamflow for the Columbia River occurs on 1 June for the historical period (1986-2005), and 19
May and 25 February, respectively, for RCP 4.5 and 8.5 scenario for the period 2081-2100. For
the Fraser River, the mean simulated peak streamflow occurs on 5 June for the historical period
(1986-2005), and 26 May and 21 April, respectively, for RCP 4.5 and 8.5 scenario for the period
2081-2100. The pronounced changes in the Fraser River basin peak flow are apparent in its flow
duration curve (Figure 10h) which shows a decrease (increase) in the frequency of streamflow
events which occurred less (more) than about 16% of the time and result in a more equitable
streamflow regime with a pronounced reduction in its seasonality. The simulated streamflow for
the St. Lawrence River shows very little seasonality and since annual streamflow increases for
both scenarios, the flow duration curve simply moves up (Figure 10j).
**3.3 Uncertainty in simulated changes in future streamflow**

Using the large ensemble simulations that are available for the historical period and the

RCP 8.5 scenario at 0.44 ° resolution we quantified the uncertainty in the simulated streamflow
associated with the internal variability of the CanRCM4 model. Similar to the 0.22° resolution,
we regridded the 0.44° runoff at CanRCM4's rotated latitude longitude projection to 0.5° regular
latitude longitude projection for use as input into the river routing scheme. This is illustrated in
Figure 11 which shows the simulated daily streamflow for all the rivers considered here except
the Nelson River. In Figure 11, the solid lines show the average across the 50 members of the
large ensemble, light shading shows the full range of the results, and dark shading shows the
mean ± one standard deviation range (this implies the 16%-84%, i.e. 68%, range when assuming
normally distributed monthly streamflow values). In addition, streamflow from the 0.22°
simulations (from Figure 10) is shown as dashed lines to allow direct comparison of results from
the 0.22° and 0.44° simulations.
The changes in simulated streamflow are consistent between the 0.22° and 0.44°
simulations. The results from the 0.44° simulations are also notably smoother compared to the
0.22° simulations since the 0.44° results are also averaged over the 50 ensemble members in
addition to the 20 years. For the most part, the results from the 0.22° simulations lie within the
full          range          of          results          from          the          0.44°          simulations.
This is expected since the driving climate at the boundaries of CanRCM4 based on CanESM2 is
the same in both resolutions. The magnitude of change from the historical to the RCP 8.5 scenario
(see legend for individual rivers) is, however, somewhat different. This is also expected because
the coarser resolution  0.44° simulations are less representative of the basin topography than
the 0.22°simulations. The day of peak streamflow occurs a few days earlier in 0.22° simulations
than in the 0.44° simulations for the Mackenzie and Yukon Rivers. Overall, the large ensemble
from the 0.44° simulations helps to provide context for results from the 0.22° simulations.
Overall, despite the differences in the magnitude of changes, the direction and variability
of change obtained from this study is generally consistent with the previous studies using basin-
scale hydrologic models, driven by statistically downscaled and bias-corrected climate model
data, for instance for the Fraser River (Islam et al., 2019; Shrestha et al., 2012), the Columbia
River (Schnorbus et al., 2014) and the Yukon River (Hay and McCabe, 2010). The results presented
here are also comparable to the projections from global and regional scale hydrologic models,
e.g. for the Mackenzie River basin (Krysanova et al., 2017, 2020).
**4. Summary and conclusions**
This study offers a consistent analysis of results across six river basins in Canada based on
results from the CanRCM4 model. Despite the biases in simulated present-day CanRCM4 climate,
and some differences in the results based on 0.22° and 0.44° simulations, the results provide
useful information about changes in simulated streamflow that is consistent with expectations
of process behaviour in a warmer climate, and with published studies.
Neither future precipitation nor temperature changes are uniform across Canada.
Simulated precipitation increases are higher closer to the west and east coasts, and simulated
temperature changes are higher towards the Arctic. Similar to precipitation, runoff changes are
also higher closer to the west and east coasts. The changes in simulated streamflow indicate how
the present-day climate state of river basins plays a role in their response to climate change. The
results yield two broadly distinct responses of monthly streamflow changes to climate warming,
up until the end of this century, for the northerly Mackenzie and Yukon rivers and the southerly
Fraser and Columbia rivers. Despite higher future projected temperature changes in Canada's
north, peak streamflow for the Mackenzie and Yukon rivers is still dominated by the spring
snowmelt. This is because the present-day colder states of these river basins imply that even
after around 6-7 °C warming, the basin-wide average temperatures are cold enough to not
sufficiently change their snowmelt-dominated streamflow regimes. Changes, however, do occur
in streamflow seasonality for these two rivers. Mean peak daily streamflow occurs earlier by
about 6-7 days for the Mackenzie and Yukon Rivers in the RCP 4.5 scenario and about 28 days for
the Mackenzie River and 12 days for the Yukon River for the RCP 8.5 scenario (Figure 10). The
earlier start of the snowmelt is the primary factor for the changes in peak streamflow and its time
of occurrence, while the streamflow increases during the rest of the year  (except for June and
part of July) are driven by an increase in precipitation. Additionally, a higher fraction of winter
precipitation occurring as rainfall drives the winter streamflow increases. In contrast, the
streamflow seasonality for the southerly Fraser and Columbia rivers is significantly more affected
by warmer temperatures because the mean annual basin-wide temperature for these river basins
is already above 0° C for the historical period. Both these rivers experience pronounced changes
in their streamflow seasonality. The peak daily streamflow for both rivers decreases considerably
and occurs about 45 days earlier for the Fraser River and about 100 days earlier for the Columbia
River in the RCP 8.5 scenario. These results compare reasonably to the 1-2 months earlier peak
in previous studies for the Fraser River (Islam et al., 2019; Shrestha et al., 2012) but are higher
than the two months earlier peak for the Columbia River (Schnorbus et al., 2014) that used results
from multiple climate models. Shrestha et al. (2021a) used CanRCM4 data to evaluate snowpack
response to varying degrees of warming. They found that snowpack reduction using CanRCM4-
LE is higher than the ensemble of results obtained by driving a hydrological model with data from
other climate models (their supplementary information), consistent with CanESM2's higher
climate sensitivity. For the Nelson and the St. Lawrence Rivers which show very little seasonality
the effect of climate change is reflected in the changes in mean annual streamflow.

The results presented here also appear to show that the simulated changes in streamflow

are somewhat resolution-dependent. This would be expected especially for topography-
dominated river basins. If a large ensemble of 50 members for the 0.22° resolution was also
available, it would have been easier to draw firm conclusions about the effect of the spatial
resolution on changes in simulated streamflow.
There are two primary limitations of the work presented here. First, we use results from
only one climate model. It would have been ideal to use runoff from other regional climate
models to provide an uncertainty range based on the spread across different climate models.
This would have also allowed us to evaluate how the spread across models compares to the
spread across the 50 members of the CanRCM4 large ensemble. Second, the results are based on
direct output from the CanRCM4 climate model and direct climate model output is biased. This
limitation is tied to our methodology. The use of bias-corrected climate data inevitably implies
using a different hydrological model or land surface scheme, than the land surface component of
CanRCM4, and forcing it with bias-corrected climate data to obtain runoff. Finally, there are
uncertainties associated with the routing process itself. As mentioned earlier, the routing scheme
accounts for ice jams in a simplified manner, and anthropogenic flow regulation is not taken into
account. The implicit assumption when using raw climate model output is that, despite the biases
in simulated climate, it is possible to derive useful information about the impact of climate
change on the simulated streamflow and other components of the hydrological budget. The
Canada-wide results presented here have allowed us to differentiate between the hydrological
response of the northerly Mackenzie and Yukon Rivers, and the southerly Fraser and Columbia
Rivers, to climate change in a consistent manner. Furthermore, our results help fill the gaps in
regions across Canada, where no climate model-driven hydrological projections are available.
Within the scope of this study, we have only evaluated streamflow at the mouth of the six major
rivers considered here. The full data set of daily simulated streamflow for the 20-year historical
(1986-2005) and future periods (2081-2100) for the two scenarios, based on runoff from the
$0.22°$ simulations, is made available as detailed in the data availability section.
Large ensembles are now becoming more common. The challenge for similar future
studies is to consider the inter-model and intra-model (based on ensemble members of the same
model) spreads in the same framework to derive an uncertainty estimate that takes into account
both types of uncertainties.

**Acknowledgment**
We thank Daniel Peters for the helpful discussions at the beginning of this work and Sal Curasi
and Gesa Meyer for providing comments on the final version of this manuscript. We also
acknowledge the efforts of the climate modelling team at the Canadian Centre for Climate
Modelling and Analysis (CCCma) who made the results from CanRCM4 available. We also thank
the two anonymous reviewers who provided useful comments and helped us address the
questions related to model bias and the differences in land surface and hydrological models.
Finally, we would like to thank our handling editor (Alexander Gruber) for taking on our
manuscript and giving us the opportunity to revise our manuscript.
**Data availability**
The CanRCM4 data from $0.22°$ simulations used in this study are available from CCCma website
(https://climate-modelling.canada.ca/climatemodeldata/canrcm/CanRCM4/). The data from
the $0.44°$ CanRCM4 large ensemble are available from Environment and Climate Change
Canada (https://open.canada.ca/data/en/dataset/83aa1b18-6616-405e-9bce-af7ef8c2031c).
NetCDF files of simulated daily streamflow from the historical (1986-2005) and the two future
scenarios (RCP 4.5 and 8.5, 2081-2100) at $0.5°$ resolution are available on Zenodo for the entire
North American domain of CanRCM4 (doi:10.5281/zenodo.12775139,
https://zenodo.org/records/12775139). These streamflow data correspond to the runoff from
the $0.22°$ simulations.
**Author contributions**
VKA designed the study and wrote the majority of the manuscript. AL implemented river
routing to operate at $0.5°$ resolution and performed all the simulations. RS and AL contributed
to the manuscript text. RS also performed a literature review of existing studies that focus on
the impact of climate change on Canadian rivers.

**Competing interests**

The authors declare that they have no competing interests.

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
