# Peer review of "The effect of climate change on the simulated streamflow of six Canadian rivers based on the CanRCM4 regional climate model"

_EGUsphere, 2024_

## Author Response (AR1)

Dear handling editor,

Thank you for giving us the opportunity to revise our manuscript. We have taken into account all of the reviewers' comments in revising our manuscript as we noted at the time in our detailed response in the discussion forum. In the text below, we note the reviewers' comments in bold and highlight the changes that we made. In addition, we note the page number where the changes were made.

**Anonymous Referee #1**

***The paper discusses the two methods used to conduct these sorts of studies and selects one approach without substantiating the claim that it will give robust results despite the biases shown in the paper regarding the main climate variables (temperature and precipitation) as well as runoff.***

As noted in our reply in the discussion forum we do not intend to advocate the use of one approach over the other one by saying the selected approach is superior. Our submitted manuscript highlighted the limitations of both approaches (using raw data from climate model and using bias-corrected data to drive hydrological models offline) in the introductory section. We have strengthened our arguments around the ability of raw climate results to draw meaningful information about climate change impacts. This is done in the summary and conclusions section (page 44 of the manuscript with tracked changes).

***The approach or routing climate model runoff is not new as it has been around for a few decades and the authors do not develop it further to include anthropogenic effects of regulation, nor natural effects of lakes and river ice processes except in a very simplistic way.***

We now explicitly note the anthropogenic effects of reservoirs and natural effect of lakes are not important for the Nelson and St. Lawrence Rivers because their streamflow shows very little seasonality. As a result, the effect of climate change is reflected in the changes in mean annual streamflow. This is done on page 43 of the manuscript with tracked changes.

***There is no attempts either to calibrate it to improve the match between observed and simulation streamflow.***

This argument follows directly from use of models that are driven offline (i.e. not within the framework of climate models) and typically driven with bias-corrected climate data.

We have now included a full paragraph to highlight the differences between hydrological and

land surface models and why it is not possible to tune the parameters of a land surface model for individual grid cells or for a region (e.g. a river basin) to reproduce a small subset of model outputs. This is done on pages 5 and 6 of the manuscript with tracked changes.

*The authors also claim that their results would be useful for adaption but adaptation is usually done at much smaller scales than presented and the changes to the flow regimes at the mouth of a large river basin are not indicative of changes occurring at the sub-basin level of those rivers which are known to be heterogenous (e.g. the Mackenzie western headwaters are very different from the eastern Canadian shield tributaries, the Nelson headwaters are mountainous but then it flows through the prairies where usual routing fails) .*

At noted in our reply to reviewers' comments in the discussion forum, it seems that the following sentence …

"Despite this, however, the response to climate change is relatively robust and there is useful information in the simulated change that can be used to inform adaptation measures."

… in our submitted manuscript was the source of this comment. We have modified this sentence as follows …

Despite this, the approach can effectively capture the effects of climate change including increased evaporative demand (Winter and Eltahir, 2012), reduced snowpack (Sala the et al., 2010; Shrestha et al., 2021a), increased winter streamflow, and earlier snowmelt-driven peak flow (L. Sushama et al., 2006; Poitras et al., 2011).

… on page 4 of the manuscript with tracked changes.

*The only positive about the paper is that it uses a consistent set of forcing data from CanRCM4 to drive the routing model. However, using a monthly time step for presenting the results does create artefacts for hydrograph time shifts and peaks as the precision is months, not days or weeks. Overall. I do not see any novelty in the study to qualify it for publishing as a research paper. It can be a climate change impact study report.*

We have revised Figures 10 and 11 (page 36 and 37 of the manuscript with tracked changes) and they are now based on daily streamflow. We have also made the fully daily streamflow data available on Zenodo as mentioned on page 45 of the manuscript with tracked changes.

*The paper overlooks the availability of a bias-corrected and downscaled dataset for the CanRCM4 model which can be used to drive the model, at least to assess whether the validity of the bold claim of robustness made at the start. This dataset covers most of the North American domain and can be found at:  https://doi.org/10.20383/103.0622 and is described and partly assessed for the Mackenzie River Basin in Asong et al. (2020) (https://doi.org/10.5194/essd-12-629-2020).*

We addressed this question in our reply to the reviewers' comments in the discussion forum.

We do not overlook that bias-corrected data are available. In fact, we explicitly mentioned the use of bias corrected data by the impacts community in the original submission (top of page 5 of the manuscript with tracked changes) including the references to these data. We do not use bias-corrected data because our approach is based on using raw climate model data.

*Eq 2 lacks units, and S (which the channel slope) is not defined and contradicts the use of "S" on Figure 2. No mention was made on how S was obtained and how the equation is solved to get both the depth and velocity of flow.*

We have included additional description and equations to describe the variable velocity flow routing algorithm (on pages 13, 14, and 15 of the manuscript with tracked changes) which makes it easy to understand how depth and flow velocity are obtained. Slope is now represented by "n" and surface storage by "S".

*Using the mean +/- 1 std dev gives a confidence interval of 0.68 not 0.90 as suggested on line 652 - assuming the ensemble results is normally distributed - another assumption that was not substantiated.*

The confidence range in Figure 11 is now based on +/- 1 std dev range and we made it clear that it corresponds to 68% confidence range.

**Anonymous Referee #2**

*This paper suggested an approach that the outputs of climate models can be directly incorporated into climate change impact assessment at a basin scale. Although the authors addressed pros and cons on two approaches which has been usually employed for climate change impact assessment, there are main concerns for publication.*

*Mismatch of spatial resolution between CanRCM4 and river network: CanRCM4 provides outputs at 0.22° and 0.44° while river networks are from the TRIP dataset at 0.5°, which indicating the spatial resolution of runoff outputs were downgraded to the river networks. The main advantage of the approach employed in this study is to directly incorporate the outputs of climate models into impact studies at a watershed scale. However, this study downgraded the spatial resolution by interpolating outputs. As the river networks can be defined by topographical characteristics usually obtained from a digital elevation system (DEM), I recommend to newly define river flow directions at each grid cell of CanRCM4 to preserve the spatial resolution at 0.22° and 0.44° as well as to avoid additional biases from spatial interpolation.*

As mentioned in our reply in the discussion forum we have added an additional paragraph citing the Arora et al. (2001) study titled "Scaling aspects of river flow routing" to illustrate that

routing at different spatial resolutions produces very similar results and the spatial scale of routing is not a significant source of bias. This aspect is discussed on page 11 of the revised manuscript with tracked changes. Page 11 also mentioned that upscaling river networks is not a trivial task.

*Regridding runoff: I do not agree that regridding runoff is straight forward as the responses (i.e., runoff) to climate forcing are depending on hydrologic characteristics (e.g., slope, soil, land cover, etc.) averaged within a grid cell. In other words, different resolutions will induce different responses (runoff).*

This comment was related to the previous comment and the additional information about scale independence of routing addresses this comment.

*Accuracy in primary variables and streamflow: From Figure 4, a considerable systematic bias in precipitation and streamflow was found, especially winter and spring flows at Columbia and Fraser, which may be from a combination of biases in climate forcing and a poorly calibrated parameter set of CLASS. Thus, the bias may bring a question to me how reliable the climate change impact assessment is. I was wondering if a bias-correction method can be applied to the routed streamflow as a post process to improve the reliability of impact assessment.*

This comment from reviewer #2, like reviewer #1, also saw our results from a hydrology lens as we mentioned in our response in the discussion forum. The additional discussion on pages 5 and 6 address the calibration within the context of hydrological and land surface models. As we mentioned in our response in the discussion forum it does not make any physical sense to apply bias-correction to the routed streamflow.

*The approach suggested in this study can be applied for only an existing river network (TRIP at 0.5°). It would be great if the authors suggest a general approach that can be applied to any resolutions (e.g., sub-watershed scales) by directly incorporating the outputs of CanRCM4.*

As mentioned in the discussion forum, we are unclear what this comment implies. When analyzing streamflow, the spatial resolution of model that generates runoff and that performs the routing may or may not be the same. The reason for choosing a 0.5-degree river network in this study was to be able to use runoff from both CanRCM resolutions given the cumbersome task of upscaling river directions to a rotated latitude longitude projection and the results from the Arora et al. (2001) study which indicates that routing is broadly insensitive to spatial resolution.

*Novelty and contribution: I appreciate it if the authors prominently address the novelty and contributions of this study.*

The novelty of our study is that it provides a consistent view of hydrological changes across Canada in a single study. While climate change impact studies are available for individual river basins, we are not aware of any Canada-wide study. This is mentioned explicitly on lines 301 to 304 on page 7 of the revised manuscript with tracked changes. We also highlight this in our abstract and in the summary and conclusions section by explicitly stating this study provides "a consistent analysis" of future climatic changes across Canada.

***Specific comments***

***Line 68-70: Climate model driven climate signals need to be preserved. However, it is questionable that the information can be used to inform adaptation measures.***

As noted in response to reviewer # 1 this was a generalized statement.

We have modified the relevant sentence as follows …

Despite this, the approach can effectively capture the effects of climate change including increased evaporative demand (Winter and Eltahir, 2012), reduced snowpack (Salathé et al., 2010; Shrestha et al., 2021a), increased winter streamflow, and earlier snowmelt-driven peak flow (L. Sushama et al., 2006; Poitras et al., 2011).

… on page 4 of the manuscript with tracked changes.

***Line 84-88: There are bias-correction methodologies that preserve GCM-driven climate signals.***

The intent here was to indicate that just like climate models have their distinct biases, bias-correction methodologies have their limitations as well. The limitations of bias correction and downscaling are noted on pages 4 and 5 of the introductory section.

***Line 295-296: The Nelson River is affected by heavily regulated flow. Thus, I do not think that this basin is a good testbed where the approach can be applied unless a set of naturalized flows can be employed.***

Yes, we agree. As we noted in our response to reviewer #1 and in our reply in the discussion forum, for the Nelson and the St. Lawrence Rivers future change in mean annual quantities are of interest to the hydroelectricity companies, which this manuscript provides. We now note this explicitly on page 43 of the revised manuscript with tracked changes.

***Line 669-672: If we do not have to look at the magnitude of changes, without routing processes, simply aggregating the runoff within a basin seems to be enough to evaluate climate change impacts.***

This comment also appears to be related to earlier comments raising doubts on whether runoff can be routed after regridding. We have added an additional paragraph on page 11 citing the Arora et al. (2001) study titled "Scaling aspects of river flow routing" to illustrate that routing at different spatial resolutions produces very similar results and the spatial scale of routing is not a significant source of bias.

The following is the list of primary changes we have made.

- We have included the discussion about the minimal effect of the spatial scale at which routing is performed has been added.
- We have included the discussion about land surface models versus hydrological models has been added in the context of their calibration for a selected or a small subset of quantities.
- We have made it clear that given the lack of seasonality in streamflow for the Nelson and the St. Lawrence rivers, climate change information lies in the simulated changes in their mean annual streamflow values.
- Figures 10 and 11 are now based on daily streamflow.
- Daily streamflow data at 0.5° resolution based on 0.22° CanRCM4's historical, and RCP 4.5 and 8.5, simulations is made available via the Zenodo website and a DOI.
- The description of the routing model is extended to include all equations which makes it clear how velocity and flow depth are modelled as prognostic variables.
- We have extended the discussion around the uncertainty related to biases in raw climate model output.

Best regards,
Authors

---

## Author Response (AR2)

*Dear handling editor,*

*Thank you for giving us the opportunity to revise our manuscript following the second round of reviews. Your three final remarks are addressed below. Our response is shown in italics and in blue colour.*

1) Referee #2 still has made two remarks. You can choose whether or not you want to add a comment addressing those into the final manuscript.

The authors addressed well my comments and suggestions in the revised manuscript except for bias-correction of streamflow and scaling aspects of river flow routing.

1) Given the observed streamflow at the hydrometric stations, i.e., outlet of each river basin, the routed streamflow can be bias-corrected as a post-process to enhance reliability in projected streamflow without calibrating hydrologic and land-surface models.

2) If spatial resolution in routing is not sensitive to the accuracy in streamflow for a large-scale basin (e.g., Mackenzie River basin), as addressed in the revised manuscript, I still think that a simple aggregation of runoff at each time step within a basin may provide streamflow comparable to those from a routing model. For example, Li et al. (2019) calculated streamflow using the direct aggregation of the runoff and baseflow over the drainage area for each gage as they confirmed that runoff routing vs aggregation differences in both streamflow timing and magnitude at the daily time scale are modest.

*We have tried our best to address all reviewers' comments from the past set of reviews, and we are glad that this reviewer is satisfied with our revisions. However, we have decided to not take into account suggestions made in this second round of review because of the reasons discussed below.*

*First, the bias correction of routed streamflow after-the-fact will not only compensate for routing biases but also biases in CanRCM4 climate, and the biases in the land surface model itself. In addition, streamflow bias correction adapted for the present day cannot be used for bias correction in the future especially for high emissions scenarios. This is because bias correction for the present day climate will not be valid for simulated future streamflow because of the shift in the timing of the peak flow, projected for all basins, and for the shift in flow regime from snow-dominated to hybrid regimes, projected for the Fraser and Columbia basins. We do not think that bias correction for routed streamflow will be scientifically defendable.*

*Second, aggregating runoff (without routing) as in Li et al. (2019) is not appropriate for our study because the catchments considered in that study are much smaller in size than the*

*continental scale river basins considered in our study. Based on our tests (not shown) the time difference in the peak runoff and peak streamflow for the Mackenzie River basin, i.e. the delay caused by routing, is of the order of about a month. This is a not a trivial delay. So in our opinion, the suggestion to not route runoff is worse than ignoring the effects of anthropogenic regulation (that we have tried to address for the Columbia River by using naturalized streamflow) and the simple treatment of ice jams in our routing approach.*

2) Line 117 in the revised manuscript with track changes: I believe "that" should be removed.

*We have broken this long sentence into two sentence and reworded it for clarity on Page 6 of the revised manuscript.*

3) I appreciated the added new discussion in Lines 116-153 and 154-162 (again in the track-changes manuscript), but it could be greatly strengthened by adding some references. Especially after the discussions during the review process, I strongly encourage you to try and find some papers that could back up these statements (if available).

*Thank you for this suggestion. We have added a couple of small additional sentences to this paragraph and four new references. This modification is also on page 6.*

Best regards,
Authors

[revised manuscript text omitted]